# Closing the Gap between Neural Networks for Approximate and Rigorous Logical Reasoning

## Abstract

Despite the historical successes of neural networks, the rigour of logical reasoning is still beyond their reach. Taking syllogistic reasoning as a subset of logical reasoning, we show supervised neural networks cannot reach the rigour of syllogistic reasoning, mainly because they use composition tables, which are coarse to distinguish each valid type of syllogistic reasoning and because end-to-end supervised learning may change the premises. As Transformer's Key-Query-Value structure is a combination table, we conclude that neural networks built upon Transformers cannot reach the rigour of syllogistic reasoning and, thus, cannot reach the rigour of logical reasoning. We logically prove that oversmoothing, in the setting of part-whole relations, can be avoided, if neural networks use region embeddings, and propose the method of reasoning through explicit constructing and inspecting region configurations, to achieve the rigour of logical reasoning.

## 1 Introduction

The power of neural networks is witnessed by its superior performance in simulating a variety of human behaviours, for example, human-like question-answering (Biever, 2023), playing games (Silver et al., 2017; Schrittwieser et al., 2020), predicting gene structures (Abramson et al., 2024), helping mathematical discovery (Davies et al., 2021), solving IMO tasks (Trinh et al., 2024). The success of neural networks may ascribe to their strong power to represent closed-world tasks (e.g., Chess, Go). RNNs and Transformers are even proven to be Turing complete (given unbounded time) (Nowak et al., 2023; Strobl et al., 2024).

However, accompanied by these exciting successes are LLMs' unpredictable behaviours (Park et al., 2024), errors in simple abstract reasoning (Eisape et al., 2024; Lampinen et al., 2024), and the irrationality of making correct answers with incorrect explanations (Creswell et al., 2022; Zelikman et al., 2022). This brings unpredictable risks to our society (Bengio et al., 2024), for example, in clinical decision-making (Hager et al., 2024). Although breaking a complex reasoning task into multiple steps may improve their reasoning performance (Creswell et al., 2022; Wei et al., 2023; Lightman et al., 2023), it remains unclear whether they reason at all (Biever, 2023; Melanie, 2023) and how far can neural networks go with the scaling law (Kaplan et al., 2020). Neural networks are invented to simulate the functions of our minds. The design is inspired by the biological structure of neurons (Anderson, 1995). Here, we show that supervised neural networks cannot achieve the rigour of syllogistic reasoning and thus cannot achieve the rigour of logical reasoning. We further show, that to achieve the rigour, neural networks shall promote vector embeddings to region embeddings and adopt the method of reasoning through model construction and inspection (Johnson-Laird & Byrne, 1991; Knauff et al., 2003; Goodwin & Johnson-Laird, 2005; Knauff, 2009), as demonstrated in (Dong et al., 2024).

The supervised learning syllogistic reasoning falls in the paradigm of statistic learning, with the stable-world assumption that the training data and the testing data share the same distribution (Goyal & Bengio, 2022; Gigerenzer, 2022). A well-trained neural network for syllogistic reasoning may reach 100% confidence for the reasoning *All Greeks are human. All humans are mortal. ∴ All Greeks are mortal.*, but will be less confident, if we replace *Greeks* with *Dani people*, or a *Papuan ethnic group*. Experiments also show that LLMs may mimic syllogistic errors in the training data,

though they can perform very well with syllogistic reasoning (Lampinen et al., 2024). Thus, using symbolic terms, neural networks cannot achieve the rigour of syllogistic reasoning. Considering the finite set-theoretic relations behind all syllogistic statements, we may first translate a syllogistic statement into a set-theoretic relation that can be represented by Euler diagram. In this way, Greeks, Dani people, and a Papuan ethnic group will be equally represented by a circle.

In Section 2, we introduce syllogistic reasoning and the criteria of rigorous syllogistic reasoning for neural networks as follows: (1) being able to correctly reason with out-of-distribution data, (2) being able to correctly reason with all valid syllogistic reasoning, and (3) not changing premises. In Section 3, we visit Euler Net, a supervised neural network that learns syllogistic reasoning by utilising a combination table of its set-theoretic semantics and achieves 99.8% accuracy. However, the performance drops to 56%, when we fed with new randomly generated data. In Section 4, we introduce a method that automatically identifies and removes out-of-distribution data. We conclude that given the reasoning types as represented by the combination table, increasing training data will improve the reasoning performance.

In Section 5, we show that Euler Net's composition table cannot distinguish all valid types of syllogistic reasoning and, thus, cannot help Euler Net to achieve the rigour of syllogistic reasoning. We improve the composition table and propose to separate the end-to-end mapping between premises (network inputs) and conclusions (network outputs) into two processes: firstly, constructing a latent model from inputs, and secondly, inspecting this model to draw conclusions. Though with sufficient data, the improved version of Euler Net may achieve nearly 100% accuracy for all valid types of syllogistic reasoning, it may need an unbounded number of inspecting models and may draw incorrect conclusions by automatically changing premises.

In Section 6, we introduce a contradictory design feature for object recognition and logical reasoning. It is a desirable feature for object recognition networks that they can recognise objects only by observing parts; however, it is an undesirable feature for reasoning networks that they will insert new concepts into the premises. Thus, if an end-to-end supervised neural network for reasoning uses supervised neural networks for recognising inputs, this reasoning neural network will not achieve the rigour of logical reasoning. Because its object recognition networks may automatically insert new concepts by recovering the whole from parts. We demonstrate this phenomenon through experiments with Euler Net and show that this problem can be relieved by defining unintended inputs and creating the training data, but cannot be solved entirely.

Syllogistic reasoning is the microcosm of human rationality. Neural reasoning for syllogistic relations are embedded in scenarios with complex objects that needs sophisticated neural architectures, e.g. Visual Transformers (Dosovitskiy et al., 2021). In Section 7, we prove that if all the outputs of neural networks oversmooth (the same embedding), the output embedding will be a single point. The contraposition of this statement is that if the output embedding is not a point, neural networks will not oversmooth. The simplest region embedding would be sphere embeddings that can be obtained by adding a non-zero radius to a vector embedding. We briefly visit Sphere Neural Network (Dong et al., 2024) and illustrate the method of reasoning through explicitly constructing and inspecting sphere configurations as Euler diagrams, and achieves the rigour of syllogistic reasoning.

## 2 SYLLOGISTIC REASONING

Aristotelian syllogistic reasoning is a logical deduction with the form of two premises and one conclusion. A syllogistic deduction only contains three terms (*Subject*, *Middle*, and *Predicate*) and four possible relations, as follows.

- *universal affirmative*: all $X$ are $Y$;
- *particular affirmative*: some $X$ are $Y$;
- *universal negative*: no $X$ are $Y$;
- *particular negative*: some $X$ are not $Y$.

For example, two premises can be *some lawyers are presidents. no presidents are scientists.* The logical conclusion is *some lawyers are not scientists.*, its negation is *all lawyers are scientists.*, as shown in Figure 1(e). The four relations can be interpreted through set relations in Euler diagrams,

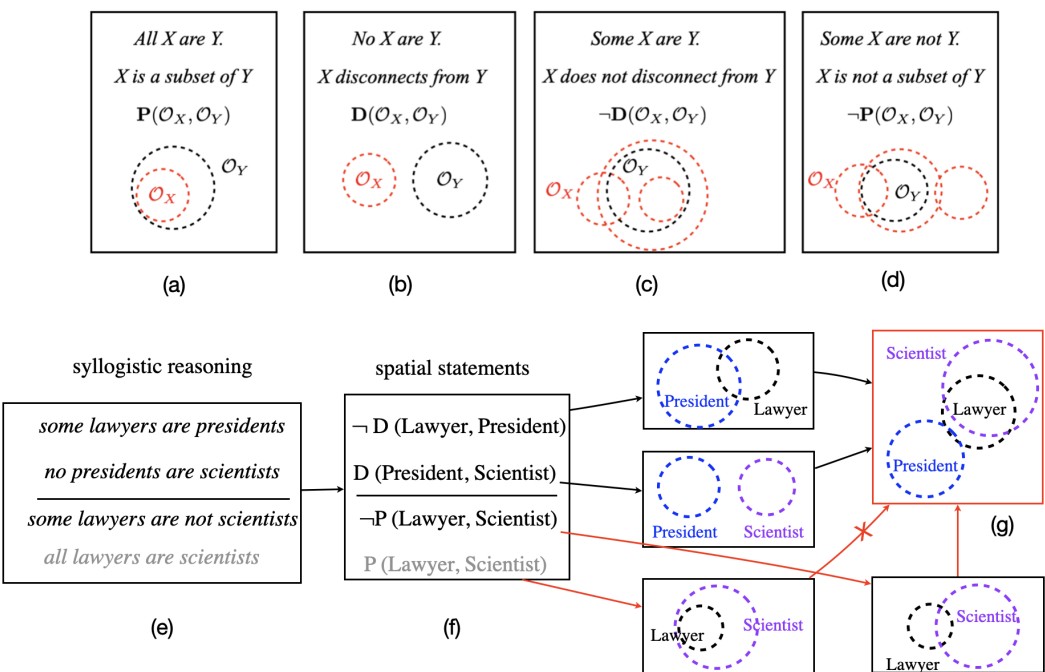

Figure 1: (a-d) Four syllogistic relations and their one-to-one mapping to spatial relations; (e) two premises are *some lawyers are presidents. no presidents are scientists*, the logical conclusion is *some lawyers are scientists*, its negation is *all lawyers are scientists*; (f) spatial statements of the syllogistic statements; (g) no sphere configuration satisfies the premises and the conclusion *all lawyers are scientists*; there is a sphere configuration that satisfies the premises and the conclusion *some lawyers are not scientists*.

as illustrated in Figure 1(a-d). For example, *some X are Y* can be interpreted as the set relation "set X does not intersect with set Y". This implies three possible relations between set X and set Y, as shown in Figure 1(c). We map a set to a sphere, e.g., mapping set X to Sphere $\mathcal{O}_X$, and translate *some X are Y* into one spatial relation: $\mathcal{O}_X$ does not disconnect from $\mathcal{O}_Y$, $\neg\mathbf{D}(\mathcal{O}_X, \mathcal{O}_Y)$. The disconnected relation $\mathbf{D}$ can be defined using the part-whole relation $\mathbf{P}$ as follows: $\mathcal{O}_X$ disconnects from $\mathcal{O}_Y$, if and only if there is no $\mathcal{O}_Z$ that is part of both $\mathcal{O}_X$ and $\mathcal{O}_Y$ (Smith, 1996).

$$\mathbf{D}(\mathcal{O}_X, \mathcal{O}_Y) \triangleq \nexists \mathcal{O}_Z \mathbf{P}(\mathcal{O}_Z, \mathcal{O}_X) \wedge \mathbf{P}(\mathcal{O}_Z, \mathcal{O}_Y)$$

In this way, we can define syllogistic relations through the part-whole relation $\mathbf{P}$ and establish a one-to-one relation ($\Leftrightarrow$) between syllogistic and spatial relations as follows.

- "all $X$ are $Y$" $\Leftrightarrow$ "sphere $\mathcal{O}_X$ is part of sphere $\mathcal{O}_Y$", $\mathbf{P}(\mathcal{O}_X, \mathcal{O}_Y)$;
- "some $X$ are $Y$" $\Leftrightarrow$ "sphere $\mathcal{O}_X$ does not disconnect from sphere $\mathcal{O}_Y$", $\neg\mathbf{D}(\mathcal{O}_X, \mathcal{O}_Y)$;
- "no $X$ are $Y$" $\Leftrightarrow$ "sphere $\mathcal{O}_X$ disconnects from sphere $\mathcal{O}_Y$", $\mathbf{D}(\mathcal{O}_X, \mathcal{O}_Y)$;
- "some $X$ are not $Y$" $\Leftrightarrow$ "sphere $\mathcal{O}_X$ is not part of sphere $\mathcal{O}_Y$", $\neg\mathbf{P}(\mathcal{O}_X, \mathcal{O}_Y)$.

A syllogistic reasoning can be satisfiable, unsatisfiable, valid, or invalid, as listed in Table 1. For example, being valid means the conclusion is true in every case its premises are true (Jeffrey, 1981); being satisfiable means there is a case that both premises and the conclusion are true. For a valid reasoning, the negation of its conclusion is unsatisfiable; for an invalid reasoning, the negation of its conclusion is satisfiable. Geometrically, a syllogistic reasoning is satisfiable, if and only if we can construct an Euler diagram, e.g., three circles satisfying the spatial relations of the premises and conclusion; otherwise, this reasoning will be unsatisfiable. In Figure 1(g), we successfully constructed an Euler diagram of the premises and the conclusion *some lawyers are not scientists*, so this reasoning is satisfiable. But, we cannot construct an Euler diagram of the premises and the conclusion *all lawyers are scientists*, so this conclusion is unsatisfiable. Therefore, its negation (*some lawyers are not scientists*) is valid.

Table 1: Four types of syllogistic reasoning.

| type | quantifier | premise | conclusion | explanation |
|---|---|---|---|---|
| valid | universal | true | true | The conclusion is true in every case premises are true (Jeffrey, 1981). |
| invalid | existential | true | false | There is a case that the premises are true and the conclusion is false. |
| satisfiable | existential | true | true | There is a case that the premises and the conclusion are true. |
| unsatisfiable | universal | true | false | The conclusion is false in every case premises are true. |

If we allow two terms in premises to change positions and fix the order of terms in the conclusion statement, there will be 256 different forms of Aristotelian syllogistic reasoning, among which 24 types are valid (Khemlani & Johnson-Laird, 2012). A reasoning network reaches the rigour of syllogistic reasoning, if it can correctly determine all valid syllogistic reasoning for all datasets in bounded computing time. This criterion requires the system to achieve 100% accuracy on out-of-distribution data or to classify them as unintended input. To be rigorous also requires the system not to change the premises. It is hard or unrealistic to develop supervised neural networks that meet these criteria. However, we may believe that supervised neural networks can infinitely close to the rigour and meet the criteria, as long as we have more and more data (the scaling law). We show that increasing the amount of data will improve reasoning performance but will not infinitely close to the rigour of logical reasoning – there is a gap, and to close this gap, we need to abandon training data and switch to the method of reasoning through model construction and inspection.

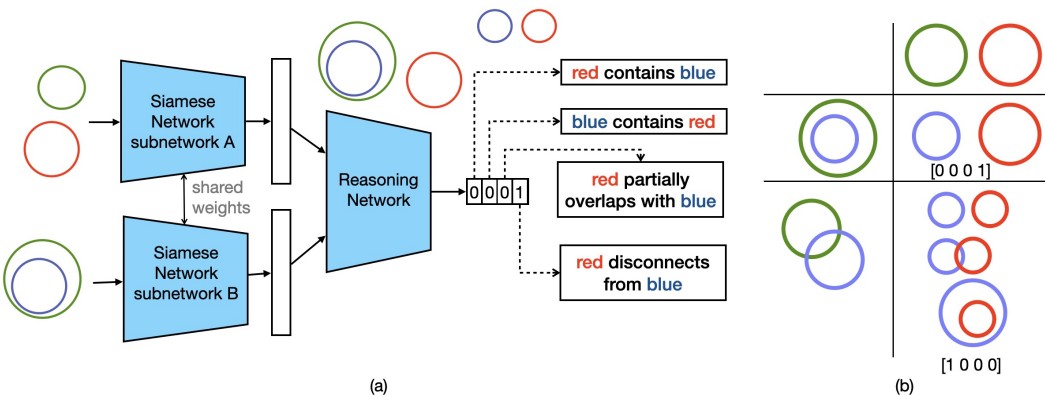

Figure 2: (a) The architecture of Euler Net; (b) The composition table of Euler Net. If the two premises are "blue is inside green. green disconnects from red", the combination result will only be "blue disconnects from red", represented as $[0, 0, 0, 1]$.

## 3 EULER NET – A SUPERVISED NEURAL REASONER FOR SYLLOGISTIC RELATIONS

Syllogistic reasoning is not a challenging task for symbolic approaches (Vukmirovic et al., 2019; Bentkamp et al., 2021), but very challenging for neural modelling and developing neural syllogistic models was once regarded a Utopian (Khemlani & Johnson-Laird, 2012). A recent comparative analysis found even the largest LLM may make mistakes in syllogistic reasoning (Eisape et al., 2024), which implicitly shows that LLMs have not achieved the rigour of syllogistic reasoning. Inspired by the structure of the human visual cortex, Wang et al. (2018; 2020) developed Euler Net, a supervised deep-learning network for syllogistic reasoning, as illustrated in Figure 2(a). The inputs of EN are two images, each consisting of two coloured circles with a set-theoretic relation. Colours of circles distinguish three terms in syllogistic reasoning. The common colour in the two inputs is the midterm. With two Siamese networks, Euler Net encodes each input image into a latent vector. The output is a vector representing the set-theoretic relation(s) between the subject and the predicate. The mapping from two premises to possible conclusions is enumerated in the combination table, where possible conclusions are symbolised as a vector, as illustrated in Figure 2(b). The structure of a piece of training data is ((image, image), vector). The benchmark dataset consists of 96000

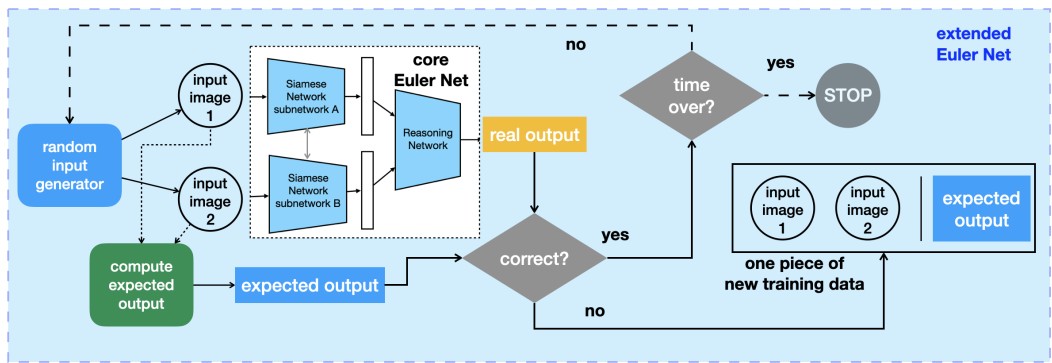

Figure 3: The random input generator creates two input images, and computes the expected output. This output is compared with the output of Euler Net. A new training record will be created, if the similarity is below a threshold.

pieces of data. They are randomly shuffled and partitioned into 80000 pieces for training, 8000 for validation, and 8000 for testing. Euler Net achieved 99.8% accuracy on the testing dataset.

## 4 AUTOMATIC IDENTIFYING AND REMOVING OUT-OF-DISTRIBUTION DATA

We fed new randomly generated test data to a well-trained Euler Net, its performance dropped from 99.8% to 56.0%. We explore whether Euler Net can approach the rigour of syllogistic reasoning. For Euler Net, it is possible to design a wrapper system that automatically identifies errors and creates new training data to improve the performance of Euler Net. The wrapper system works as follows: it firstly randomly generates geometric parameters of circles, i.e., centre points and radii. From this information, the wrapper system will create input images and the expected output vector. Then, the wrapper system feeds the input images into Euler Net and gets the output vector of Euler Net. If this vector differs to a degree from the expected vector, the pair of the input images and the expected vector is a new piece of training data (Algorithm 1). After the amount of new pieces of training data reaches a value, e.g., 10000, or a time limit is reached, the current Euler Net will be trained and tested and promoted into a new version with better performance, as illustrated in Figure 3. After 20 times of repeating the process, the Euler Net (version 20) improves the accuracy to 97.8% for randomly generated testing data. The result of this experiment supports the scaling law. We can envisage that Euler Net may infinitely close to 100% accuracy, given unbounded times of collecting new pieces of training data and unbounded times of repeating the process.

---

**Algorithm 1:** Random search for new training data

---

**Input:** Euler Net: $\mathcal{EN}$; The maximum size of unintended data set: MaxDataSize; The timer for random searching: Timer; The maximum time for searching: MaxSearchTime; The threshold to be unintended: Threshold

**Output:** A new data set for training: NewTrainingData

1 NewTrainingData $\leftarrow \emptyset$; DataSize $\leftarrow 0$; Timer $\leftarrow$ `start_timer`()
2 **while** DataSize $<$ MaxDataSize $\wedge$ Timer $<$ MaxSearchTime **do**
3      OneInput $\leftarrow$ `randomly_generate_one_input` ()
4      Output $\leftarrow$ `get_output_of_network` ($\mathcal{EN}$, OneInput)
5      COutput $\leftarrow$ `compute_correct_output` (OneInput)
6      **if** `loss`(Output, COutput) $>$ Threshold **then**
7          NewTrainingData $\leftarrow$ NewTrainingData $\cup \{($OneInput, COutput$)\}$
8          DataSize $\leftarrow$ DataSize $+ 1$

9 **return** NewTrainingData

---

Table 2: Performances of Euler Net for each valid type of syllogistic reasoning.

| Valid Type | Accuracy | Valid Type | Accuracy | Valid Type | Accuracy |
|------------|----------|------------|----------|------------|----------|
| BARBARA | 100% | BARBARI | 50% | BAROCO | 66.7% |
| BAMALIP | 50% | BOCARDO | 75% | CALEMES | 100% |
| CAMESTROS | 50% | CELARENT | 100% | CESARO | 50% |
| CALEMO | 50% | CESARE | 100% | CELARONT | 50% |
| DARAPTI | 100% | DARII | 75% | DISAMIS | 75% |
| FESAPO | 100% | DATISI | 75% | DIMATIS | 75% |
| FELAPTON | 100% | FERIO | 83.3% | FERISON | 83.3% |
| CAMESTRES | 100% | FRESISON | 83.3% | FESTINO | 83.3% |

## 5 COMPOSITION TABLE CANNOT DISTINGUISH EACH VALID TYPE

### 5.1 NEW TESTING DATA-SET

To examine whether Euler Net covers all 24 types of valid syllogistic reasoning, we created a new testing dataset. We group 24 *valid* syllogism types into 14 groups, as '*no x are y*' has the same meaning with '*no y are x*'; and *some x are y* has the same meaning with '*some y are x*'. For each group, we created 500 test cases by extracting hypernym relations from WordNet-3.0 Miller (1995), each test case consisting of one valid conclusion and its negation, totalling 14000 syllogism reasoning tasks. In the hypernym structure, *elementary_particle.n.01* is a descendent of *natural_object.n.01* and *artifact.n.01* is not a descendent of *natural_object.n.01*. So, we create the valid syllogistic reasoning as: *all elementary_particle.n.01 are natural_object.n.01*, *no artifact.n.01 are natural_object.n.01*, ∴ *no elementary_particle.n.01 are artifact.n.01*. Its negation will be : *all elementary_particle.n.01 are natural_object.n.01*, *some artifact.n.01 are natural_object.n.01*. ∴ *some elementary_particle.n.01 are artifact.n.01*

### 5.2 PERFORMANCE OF EULER NET ON THE NEW DATA-SET

We use the pre-processing tool of Euler Net to transform premises into coloured circles, and conclusions into vectors, respectively, and fed to a well-trained Euler Net. It works very well if a task falls into a valid syllogistic structure: For 8 syllogistic structures, Euler Net reaches 100% accuracy, namely, BARBARA, CELARENT, CESARE, DARAPTI, CALEMES, CAMESTRES, FELAPTON, and FESAPO. Accuracies of the rest 16 types range from 50% to 83.3%. The overall accuracy is 76%, as shown in Table 2.

### 5.3 COMBINATION TABLES ARE TOO COARSE

The reason for the performance drop is Euler Net's combination table fails to distinguish each valid type of syllogistic reasoning. The same premises can have different valid syllogistic conclusions. For example, the combination of "*V circle is inside W circle*" with "*W circle is inside U circle*" is the diagram that "*V circle is inside U circle*". This is one valid type of syllogistic reasoning (the Barbara type) "*all V are W. all W are U. ∴ all V are U.*" (each valid syllogistic reasoning has a name; we list all 24 valid syllogistic reasoning in the supplementary material). However, from the same premises, we can also conclude "*some V are U.*", which is another valid type of syllogistic reasoning, namely, the Barbari type. Though "*Some V are U.*" is weaker, it is valid, which includes three relations between *V* and *U* as follows: "*V circle is inside from U circle*", "*U circle is inside from V circle*", and "*V circle is partially overlapped with U circle*"). Usually, we call "*all V are U*" the logical conclusion and "*some V are U*" its logical consistency. The combination table only partially represents the case of logical consistency (all partially represented reasoning types are marked with gray backgrounds in Figure 4). Additionally, the combination table may need three rows or columns to represent one valid type of syllogistic reasoning. For example, "*Some V are W. All W are U. ∴ Some V are U.*" The premise "*Some V are W*" may have three different diagrammatic relations, as shown in Figure 4.

Euler Net's combination table uses four set-theoretic relations between circles to exhaust all possible combinations. It fails to establish a one-to-one mapping between syllogistic statements and Euler

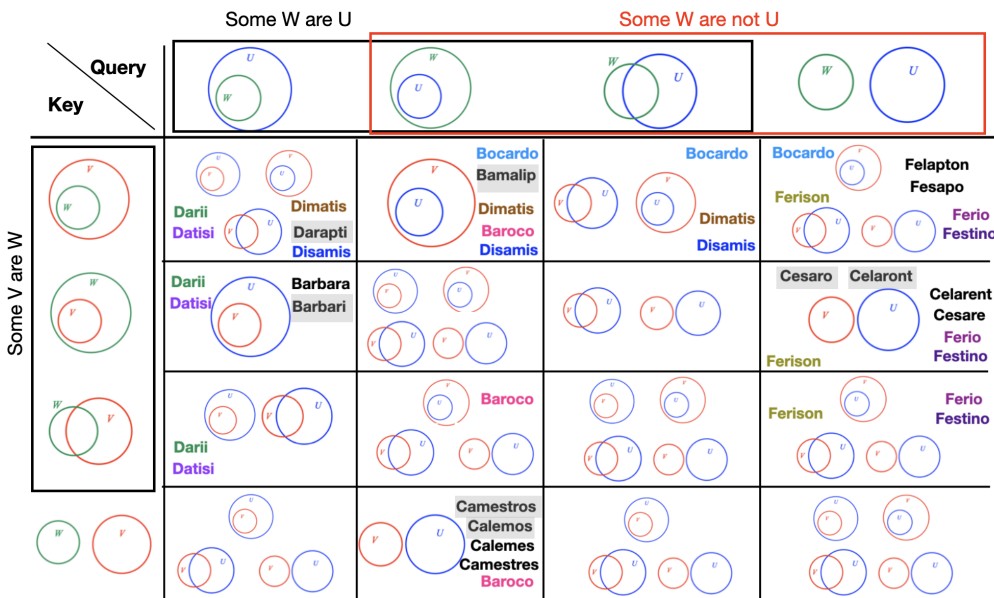

Figure 4: The combination table establishes associations between inputs (premises) and output (conclusion). It is a partial mapping to valid syllogistic reasoning. Types with grey backgrounds are partially represented. This table is also a Key-Query-Value structure.

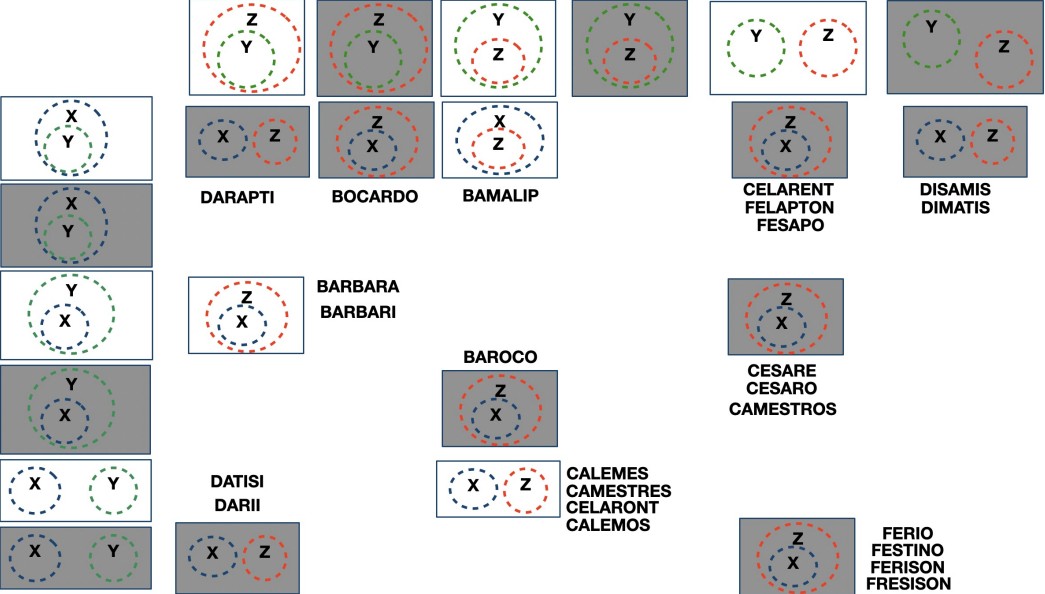

Figure 5: Each row/column corresponds to one and only one syllogistic statement. This table cannot separate logical conclusion from logical consistency.

diagrams. We propose a novel combination table, as shown in Figure 5, where we use the grey background colour to represent the negation. As "*Some X are not Y*" can be interpreted as "*it is not true that X is part of Y*", it can be mapped to one grey-background Euler diagram "*X circle is inside Y circle*". In this way, two syllogistic premises will only correspond to one cell in the combination.

However, this improved combination table still cannot separate a logical conclusion from its logical consistency. For example, "*All X are Z*" (Barbara) and "*Some X are Z*" (Barbari) share the same cell. If we solve this problem using multiple labelled outputs, each output will share a probability. In the

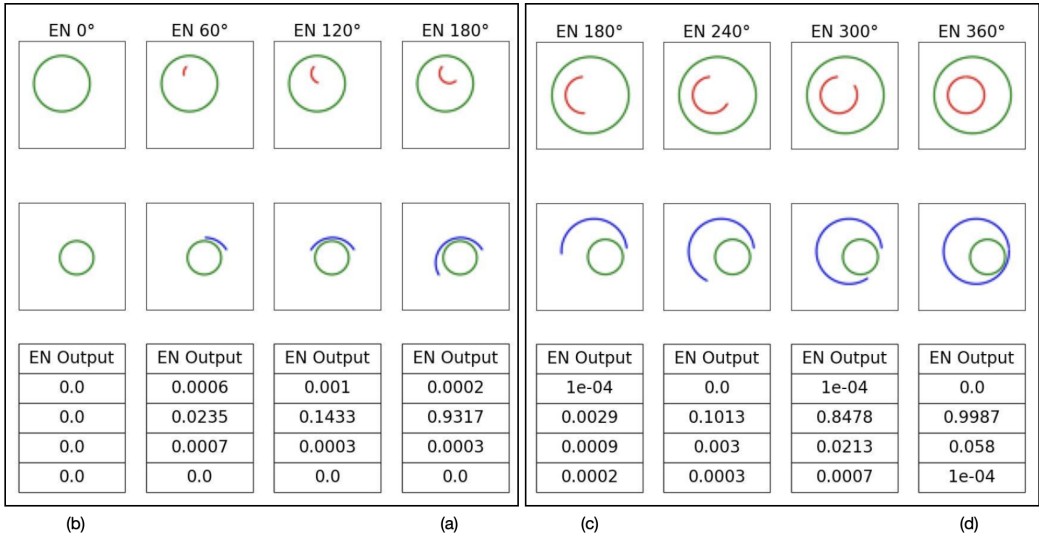

Figure 6: (a) Euler Net (EN) may automatically complete half circle into full circle (with the output $[0.0002, 0.9317, 0.0003, 0]$). As we decrease the length of the arc to $120°$, $60°$, and $0°$, EN decreases this value accordingly. (b) Euler Net (EN) may automatically ignore the half circle and only take one green circle as input (with the output $[0.0001, 0.0029, 0.0009, 0.0002]$). As we increase the length of the arc to $240°$, $300°$, and $360°$, EN increases this value accordingly.

case of Barbara and Barbari, each will have 50% confidence of being valid, which is the same as the confidence of tossing a coin. So, combination tables with multiple-labelled outputs will not achieve the rigour of logical reasoning.

## 6    END-TO-END NEURAL REASONING PROCESS MAY CHANGE PREMISES

We use the method of maximization activation  (Samek et al., 2019) to examine whether Euler Net will change premises. Concretely, we fix the output of the Euler Net and search two input images, each with only one green circle, with the target that inputs shall maximize the confidence score of the fixed output. Our experiments showed that there were input images with only one green circle that could lead Euler Net to give syllogistic conclusions between a blue circle and a red circle. For example, when each input image only has one green circle, Euler Net might output *the red circle is inside the blue circle* ($[0, 1, 0, 0]$). This is an amazing ability to predict/complete a single object, but not a desirable capability for logical reasoning, as this ability will let Euler Net change the premise of logical reasoning.

The rigour of syllogistic reasoning requires Euler Net to determine that single-circled images are invalid input. We created a new output vector $[0, 0, 0, 0]$ representing invalid inputs and trained Euler Net to be capable of classifying single-circled inputs into $[0, 0, 0, 0]$, reaching 100% accuracy. The improved Euler Net can classify single-circle inputs as invalid and perform syllogistic reasoning for normal inputs. However, this does not follow that it immediately reaches 100% accuracy for any inputs. We report our experiments to show that training data automatically creates unindented inputs as follows.

We create new datasets: one image with a green circle and a half-red circle and the other with a half-blue circle and a green circle. Our experiments show that sometimes Euler Net completes the two half circles into two whole circles, and concludes $[0, 1, 0, 0]$ *the blue circle contains the red circle*, as shown in Figure 6(a). In this case, if we decrease the arc length of the two half circles, the confidence value flagging *the blue circle contains the red circle* will decrease correspondingly, as shown in images and outputs from Figure 6(a) to (b). Sometimes, it completes one green circle and a half circle into one green circle (half circles are neglected) and concludes the inputs are invalid $[0, 0, 0, 0]$, as shown in Figure 6(c). In this case, if we increase the arc length of the two half circles,

the confidence value flagging *the blue circle contains the red circle* will increase correspondingly, as shown in images and outputs from Figure 6(c) to (d). To make Euler Net deterministic, we may define that one green circle and a half circle are invalid inputs and create enough training data to improve Euler Net. This, however, will automatically create another kind of unintended pattern: one green circle and a $(180° + 360°)/2 = 270°$ partial circle. This loop will never end.

This unfortunate fact will happen to all neural classification systems: Let $I_1$ and $I_2$ be two typical input types for two different classes $K_1$ and $K_2$, the new type of input by combining full $I_1$-type input and partial $I_2$-type input (that is an input by removing part of a standard $I_2$-type input) will be an unintended input for the well-trained neural network – its decision will switch between $K_1$ and $K_2$ in an indeterministic manner. This unfortunate fact comes along with the desirable feature of supervised object recognition that being able to recover the whole from its parts automatically.

## 7 BRIDGING THE GAP

Because combination tables are coarse and because end-to-end mapping from premises to conclusions, Euler Net cannot distinguish logical conclusions from their weaker version statements and may change premises, thus can not achieve the rigour of symbolic-level reasoning. These two limitations in representation and in methodology cause neural networks using the same or similar representation and methodology not to reach the rigour of syllogistic reasoning, for example, Transformers. With the self-attention mechanism, Transformers automatically learn Key-Query-Value structures that capture association relations between/among concepts within or across sentences, texts, or patches. A Key-Query-Value structure is a special combination table: given a Key (a row of a table) and a Query (a column of a table), the table returns a Value (a cell of the table). Thus, the best performance of Visual Transformers can reach is by learning the new combination table, as illustrated in Figure 5, with which Transformers will not achieve rigorous syllogistic reasoning. Multilayered visual transformers may be used to reason with syllogistic relations in complex objects, e.g., spatial relations among cars in street scenes. However, Transformers suffer from oversmoothing when their depth increases (outputs converge to the same feature embedding) (Park & Kim, 2022; Wang et al., 2022; Guo et al., 2023). It may be possible to avoid oversmoothing by artful reparameterization of the weights, as suggested by an ongoing research (Dovonon et al., 2024), here, we show that in the setting of part-whole relations, for any neural network that outputs region embeddings, if all outputs converge to the same feature embedding (oversmoothing), these embeddings must be a single point.

**Theorem 1.** *for any neural network that outputs region embeddings, if over-smoothing (all output feature embeddings are the same), these output embeddings will be a point.*

The proof is sketched in the supplementary material. This theorem is consistent with Wang et al. (2022)'s findings, which applied discrete Fourier analysis to ViTs and found that ViTs are low-pass filters – they lose all feature expressive power (only waves with the frequency of 0 can pass, which is a static point) after ViT networks reach 12-th depth. The contraposition of Theorem 1 is that *if the output embeddings are not points, they will not over-smooth.*

This suggests a method to completely avoid oversmoothing and, meanwhile, a necessary feature for logical reasoning, namely, neural networks using non-vector region embedding, e.g., Gaussian distribution (Athiwaratkun & Wilson, 2017), boxes (Ren et al., 2020), Beta distribution (Ren & Leskovec, 2020), cones (Zhang et al., 2021), sphere embeddings (Dong et al., 2024).

Configurations of regions may explicitly represent explicit human-like semantic representations, such as Euler diagrams in the vector space. By introducing the method of reasoning through explicit model construction and inspection (Johnson-Laird & Byrne, 1991; Knauff et al., 2003; Goodwin & Johnson-Laird, 2005; Knauff, 2009), we can separate the neural process of object recognition from the neural reasoning process. Models can be constructed without training data, as follows: The construction process initialises a simple configuration, e.g., all objects coincide, then, the configuration will be repeatedly optimised – the next model will be revised following the principle of minimal changes from the current one (Harman, 1986; Gärdenfors, 1988; Gädenfors, 1990), till either the target configuration is obtained or a stop criterion is reached, as demonstrated in the Sphere Neural Network (SphNN) (Dong et al., 2024), whose criterion can be stated as follows: for any satisfiable Aristotelian syllogistic statements, it can correctly construct a sphere configuration as Euler diagram at the global loss of zero (without using training data) in one epoch, as shown in Figure 7. So, at

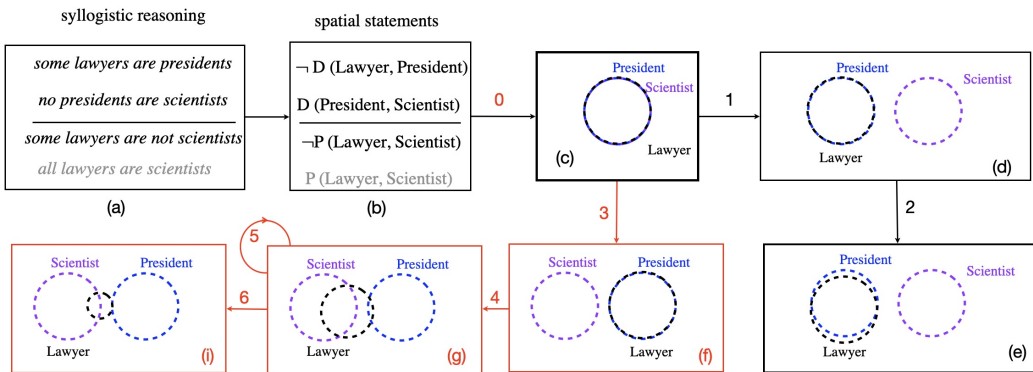

Figure 7: (a) given three syllogistic statements "some lawyers are presidents. no presidents are scientists. some lawyers are not scientists.", (b) SphNN will translate them into spatial relations between spheres as the target, then (c) initialise three coincided spheres, then (d) fix the president sphere, and move the scientist sphere away till they are disjoint, and (e) fix the president and the scientist spheres, optimise the lawyer sphere till it is not part of the president sphere, while keeping the lawyer sphere is not disjoint from the president sphere. When the global loss is zero, a model is constructed. For the unsatisfiable conclusion, "all lawyers are scientists," it is impossible to move the lawyer sphere into the scientist sphere while disconnecting from the president sphere. When the global loss stops decreasing, SphNN concludes the three statements are unsatisfiable.

the end of the first epoch, if SphNN fails to construct an Euler diagram, it will conclude the input syllogistic statements are unsatisfiable.

## 8  CONCLUSIONS AND DISCUSSIONS

By taking syllogistic reasoning, a subset of logical reasoning, we show the representation limitation of using composition tables and the method limitation of reasoning through direct mapping between premises and conclusion – the training data may not consider all situations and may only superficially capture literal meanings of training data (Bengio et al., 2024). We show that when a cell of the composition table exactly matches one valid syllogistic reasoning, the scaling law may work well. By using sphere embedding and the method of reasoning by model construction and inspection, the Sphere Neural Network achieves the rigour of syllogistic reasoning. This encourages us to re-design a new Euler Net as follows: it takes the same input as the Sphere Neural Network. Two premises are translated into Euler diagram images in the composition shown in Figure 5. Its model construction process will map the two images into the image in the table cell. Then, we train an inspection network for each valid conclusion, totalling 24 inspection networks. This new architecture may satisfy all three criteria of achieving the rigour of syllogistic reasoning: Euler diagram images, being translated by a fixed method from syllogistic statements, will follow the same distribution. So, no out-of-distribution images; Premise statements will construct one model. Each valid type will be identified by its well-trained neural network; there is no chance to change premises. Given enough training data, this new Euler Net will approach to near 100% accuracy for all syllogistic reasoning. However, given "all Greeks are human. all humans are mortal.", the inspection network that confirms "all Greeks are mortal." will deny "some Greeks are mortal." If we introduce more qualifiers into syllogistic reasoning (Khemlani, 2021), such as "most", "half", and "quarter", we must train new inspection neural networks for each. If we allow quantifiers having numbers, such as "at least three", "exactly five", "at most ten" or allow particular instances, such as "Socrates is mortal", "Aristotle is mortal", we may need an unbounded number of inspection networks. Instead of asking, to which degree it achieves syllogistic reasoning, we may doubt whether such a system truly learns syllogistic reasoning.

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
