# SUPPLEMENTARY MATERIAL

## CONTENTS

# 1 CODE AND DATASETS

Code and datasets are available at

`https://anonymous.4open.science/r/xEN-AF83/ReadMe.md.`

# 2 PROOF

We adopt definitions of the point and the extension region and three axioms of the connection relation in (Dong, 2008) for the proof.

A point is a geometric entity without extensions, which can be understood as an object that if any two objects connect with it, the two objects will connect with each other.

**Definition 1.** *A geometric object $p$ is a point, $\mathbf{Pt}(p)$, is defined as that for any geometric objects $x, y$, if $p$ connects with $x$ and $y$, then $x$ will connect with $y$.*

$$\mathbf{Pt}(p) \triangleq \forall x, y[\mathbf{C}(x, p) \wedge \mathbf{C}(y, p) \rightarrow \mathbf{C}(x, y)]$$

The connection relation is governed by the axiom as follows: if two objects connect with each other, for any third object that can be moved to a location where it connects with the first two objects.

**Axiom 1.** $\forall \mathcal{O}_1, \mathcal{O}_2[\mathbf{C}(\mathcal{O}_1, \mathcal{O}_2) \rightarrow \forall \mathbf{0} \exists \mathcal{O}[\mathcal{O} \in \mathbf{0} \wedge \mathbf{C}(\mathcal{O}_1, \mathcal{O}) \wedge \mathbf{C}(\mathcal{O}_2, \mathcal{O})]]$

Here, $\mathbf{0}$ is any $\mathcal{O}$-shaped object, and objects are only distinguished by their shapes, so $\mathbf{0}$ is also the type of $\mathcal{O}$ and we write $\mathtt{type}(\mathcal{O})$ as the type of $\mathcal{O}$. Any two objects $\mathcal{O}_1, \mathcal{O}_2$ in $\mathtt{type}(\mathcal{O})$, $\mathcal{O}_1$ can be moved and rotated to completely coincide with $\mathcal{O}_2$. Thus, $\forall \mathbf{0} \exists \mathcal{O}[\dots]$ means *for any object, it can be moved and roated to a place where this object satisfies such and such spatial relations.*

The connection relation is self-reflective and symmetric. Any object connects with itself.

**Axiom 2.** $\forall \mathcal{O}[\mathbf{C}(\mathcal{O}, \mathcal{O})]$

For any objects $\mathcal{O}_1, \mathcal{O}_2$, if $\mathcal{O}_1$ connects with $\mathcal{O}_2$, then $\mathcal{O}_2$ connects with $\mathcal{O}_1$.

**Axiom 3.** $\forall \mathcal{O}_1, \mathcal{O}_2[\mathbf{C}(\mathcal{O}_1, \mathcal{O}_2) \to \mathbf{C}(\mathcal{O}_2, \mathcal{O}_1)]$

A man with a stick can reach objects that are located further away from him. The space that the man with his stick can reach is the spatial extension of the body. Let $\mathcal{O}_m$ and $\mathcal{O}_s$ be the body of the man and the stick. $\mathcal{O}_m^{\mathcal{O}_s}$ is the extension of $\mathcal{O}_m$ by $\mathcal{O}_s$, which means any object $\mathcal{O}_x$ connecting with $\mathcal{O}_m^{\mathcal{O}_s}$, the man can move his stick to touch $\mathcal{O}_x$.

**Definition 2.** *An extension of an object $\mathcal{O}$ with an object $\mathcal{O}_e$ is the object $\mathcal{O}^{\mathcal{O}_e}$ such that any object $\mathcal{O}_x$ connects with $\mathcal{O}^{\mathcal{O}_e}$, there is an object $\mathcal{O}'_e$ that connects with $\mathcal{O}$ and $\mathcal{O}_x$, where $\mathcal{O}'_e$ and $\mathcal{O}_e$ are of the same type, written as $\mathcal{O}'_e \in \mathtt{type}(\mathcal{O}_e)$.*

$$\mathcal{O}^{\mathcal{O}_e} \triangleq \iota \mathcal{O}[\forall \mathcal{O}_x \mathbf{C}(\mathcal{O}_x, \mathcal{O}) \equiv \exists \mathcal{O}'_e[\mathcal{O}'_e \in \mathtt{type}(\mathcal{O}_e) \wedge \mathbf{C}(\mathcal{O}'_e, \mathcal{O}) \wedge \mathbf{C}(\mathcal{O}'_e, \mathcal{O}_x)]]$$

**Lemma 1.** *For any $\mathcal{O}_X$ and $\mathcal{O}_W$, if $\mathcal{O}_X$ connects with $\mathcal{O}_W$, there is $\mathcal{O}_\epsilon$ such that $\mathcal{O}_\epsilon$ is part of both $\mathcal{O}_X$ and an extension of $\mathcal{O}_W$.*

$$\forall \mathcal{O}_W, \mathcal{O}_X[\mathbf{C}(\mathcal{O}_W, \mathcal{O}_X) \to \exists \mathcal{O}_\epsilon[\mathbf{P}(\mathcal{O}_\epsilon, \mathcal{O}_X) \wedge \mathbf{P}(\mathcal{O}_\epsilon, \mathcal{O}_{W^\epsilon})]]$$

**Proof (sketch) 1.** *Let $\mathcal{O}_{W^\epsilon}$ be an extension of $\mathcal{O}_W$*

$$\mathcal{O}_{W^\epsilon} \triangleq \iota \mathcal{O}[\forall y \mathbf{C}(y, \mathcal{O}) \equiv \exists \epsilon_0[\epsilon_0 \in \mathtt{type}(\epsilon) \wedge \mathbf{C}(\epsilon_0, \mathcal{O}_W) \wedge \mathbf{C}(\epsilon_0, y)]] \qquad (1)$$

$$\frac{\mathbf{C}(\mathcal{O}_X, \mathcal{O}_W) \quad (*)}{\forall \mathtt{Z} \exists \mathcal{O}_Z[\mathcal{O}_Z \in \mathtt{Z} \wedge \mathbf{C}(\mathcal{O}_X, \mathcal{O}_Z) \wedge \mathbf{C}(\mathcal{O}_Z, \mathcal{O}_W)] \quad (**)} \text{ Axiom1}$$

$$\frac{(**) \qquad \textit{Let } \mathtt{Z} \textit{ be } \mathtt{type}(\epsilon) \quad \mathcal{O}_Z \textit{ be } \epsilon_0}{\exists \epsilon_0[\epsilon_0 \in \mathtt{type}(\epsilon) \wedge \mathbf{C}(\mathcal{O}_X, \epsilon_0) \wedge \mathbf{C}(\epsilon_0, \mathcal{O}_W)] \quad (+)}$$

$$\frac{(1) \qquad \textit{Let } y \textit{ be } \mathcal{O}_X}{\iota \mathcal{O}[\mathbf{C}(\mathcal{O}_X, \mathcal{O}) \equiv \exists \epsilon_0[\epsilon_0 \in \mathtt{type}(\epsilon) \wedge \mathbf{C}(\epsilon_0, \mathcal{O}_W) \wedge \mathbf{C}(\epsilon_0, \mathcal{O}_X)]] \quad (2)}$$

$$\frac{(2)}{\mathbf{C}(\mathcal{O}_X, \mathcal{O}_{W^\epsilon}) \equiv \exists \epsilon_0[\epsilon_0 \in \mathtt{type}(\epsilon) \wedge \mathbf{C}(\epsilon_0, \mathcal{O}_W) \wedge \mathbf{C}(\epsilon_0, \mathcal{O}_X)] \quad (3)} \textit{ the definition of } \iota$$

$$\frac{(+) \quad (3)}{\mathbf{C}(\mathcal{O}_X, \mathcal{O}_{W^\epsilon}) \quad (4)}$$

$\therefore \forall \mathcal{O}_X \mathbf{C}(\mathcal{O}_X, \mathcal{O}_W) \to \mathbf{C}(\mathcal{O}_X, \mathcal{O}_{W^\epsilon}). \quad (*)(4)$

*So, $\mathcal{O}_W$ is part of both $\mathcal{O}_{W^\epsilon}$, $\mathbf{P}(\mathcal{O}_W, \mathcal{O}_{W^\epsilon})$.*

*Let $\mathcal{O}_\epsilon$ be $\mathcal{O}_{W^\epsilon} \cap \mathcal{O}_X$. Any $\mathcal{O}_Y$, if $\mathcal{O}_Y$ connects with $\mathcal{O}_\epsilon$, $\mathcal{O}_Y$ will connect with $\mathcal{O}_{W^\epsilon}$ and $\mathcal{O}_X$. Therefore $\mathcal{O}_\epsilon$ is part of $\mathcal{O}_X$ and $\mathcal{O}_{W^\epsilon}$.*

$$\forall \mathcal{O}_W, \mathcal{O}_X[\mathbf{C}(\mathcal{O}_W, \mathcal{O}_X) \to \exists \mathcal{O}_\epsilon[\mathbf{P}(\mathcal{O}_\epsilon, \mathcal{O}_X) \wedge \mathbf{P}(\mathcal{O}_\epsilon, \mathcal{O}_{W^\epsilon})]]$$

$\square$

**Theorem 1.** *If over-smoothing (all output feature embeddings are the same), these output embeddings will be points.*

**Proof (sketch) 2.** *We consider part-whole relations among feature embeddings and for any $\mathcal{O}_V$ and $\mathcal{O}_W$, if $\mathcal{O}_V$ is a part of $\mathcal{O}_W$, $\mathcal{O}_V$ and $\mathcal{O}_W$ will coincide, $\mathbf{EQ}(\mathcal{O}_V, \mathcal{O}_W)$. We prove that $\mathcal{O}_W$ is a point $\mathbf{Pt}(\mathcal{O}_W)$.*

*For any $\mathcal{O}_X$, if $\mathcal{O}_X$ connects with $\mathcal{O}_W$, there is a region $\mathcal{O}_\epsilon$ such that $\mathcal{O}_\epsilon$ is part of $\mathcal{O}_X$ and part of an extension of $\mathcal{O}_W$ (Lemma 1), namely,*

$$\forall \mathcal{O}_W, \mathcal{O}_X[\mathbf{C}(\mathcal{O}_W, \mathcal{O}_X) \to \exists \mathcal{O}_\epsilon[\mathbf{P}(\mathcal{O}_\epsilon, \mathcal{O}_X) \wedge \mathbf{P}(\mathcal{O}_\epsilon, \mathcal{O}_{W^\epsilon})]] \qquad (5)$$

*As any feature coincides with its any part, we have $\mathbf{EQ}(\mathcal{O}_{\epsilon_0}, \mathcal{O}_{W^\epsilon})$ and $\mathbf{EQ}(\mathcal{O}_{\epsilon_0}, \mathcal{O}_X)$, so*

$$\mathbf{P}(\mathcal{O}_{W^\epsilon}, \mathcal{O}_{\epsilon_0}) \wedge \mathbf{P}(\mathcal{O}_{\epsilon_0}, \mathcal{O}_X) \qquad (6)$$

$$\frac{}{\mathbf{P}(\mathcal{O}_{W^\epsilon}, \mathcal{O}_{\epsilon_0}) \wedge \mathbf{P}(\mathcal{O}_{\epsilon_0}, \mathcal{O}_X) \rightarrow \mathbf{P}(\mathcal{O}_{W^\epsilon}, \mathcal{O}_X) \quad (7)} \ \textit{the transitive relation of } \mathbf{P}$$

$$\frac{(6) \quad (7)}{\mathbf{P}(\mathcal{O}_{W^\epsilon}, \mathcal{O}_X) \quad (8)}$$

$$\frac{(5) \quad (7)}{\forall \mathcal{O}_X \mathbf{C}(\mathcal{O}_W, \mathcal{O}_X) \rightarrow \mathbf{P}(\mathcal{O}_{W^\epsilon}, \mathcal{O}_X) \quad (8)}$$

$$\frac{(8) \quad \mathbf{P}(\mathcal{O}_{W^\epsilon}, \mathcal{O}_X) \rightarrow \mathbf{P}(\mathcal{O}_W, \mathcal{O}_X)}{\forall \mathcal{O}_X \mathbf{C}(\mathcal{O}_W, \mathcal{O}_X) \rightarrow \mathbf{P}(\mathcal{O}_W, \mathcal{O}_X) \quad (9)}$$

$$\frac{\mathbf{P}(\mathcal{O}_W, \mathcal{O}_X)}{\forall \mathcal{O}_Y [\mathbf{C}(\mathcal{O}_Y, \mathcal{O}_W) \rightarrow \mathbf{C}(\mathcal{O}_Y, \mathcal{O}_X)] \quad (10)} \ \textit{definition of } \mathbf{P}$$

$$\frac{(9) \quad (10)}{\forall \mathcal{O}_X [\mathbf{C}(\mathcal{O}_W, \mathcal{O}_X) \rightarrow \forall \mathcal{O}_Y [\mathbf{C}(\mathcal{O}_Y, \mathcal{O}_W) \rightarrow \mathbf{C}(\mathcal{O}_Y, \mathcal{O}_X)]] \quad (11)} \ \textit{rewrite } \mathbf{P}$$

*(11) is equivalent with*

$$\forall \mathcal{O}_X, \mathcal{O}_Y [\mathbf{C}(\mathcal{O}_X, \mathcal{O}_W) \wedge \mathbf{C}(\mathcal{O}_Y, \mathcal{O}_W) \rightarrow \mathbf{C}(\mathcal{O}_X, \mathcal{O}_Y)] \qquad (12)$$

$$\frac{(12)}{\mathcal{O}_W \textit{ is a point}} \ \textit{Definition of } \mathbf{Pt} \hfill \square$$

## 3 VARIOUS FORMS OF LOGICAL REASONING

We list 36 forms of logical reasoning that have the root on syllogistic reasoning (Betz et al., 2021).

### 3.1 GENERALIZED MODUS PONENS

#### 3.1.1 BASIC SCHEMA

$$\forall x F(x) \rightarrow G(x)$$
$$\frac{F(a)}{G(a) \quad \therefore}$$

#### 3.1.2 NEGATIVE VARIANT

$$\forall x F(x) \rightarrow \neg G(x)$$
$$\frac{F(a)}{\neg G(a) \quad \therefore}$$

#### 3.1.3 COMPLEX PREDICATES

$$\forall x F(x) \wedge H(x) \rightarrow G(x)$$
$$F(a)$$
$$\frac{H(a)}{G(a) \quad \therefore}$$

#### 3.1.4 DE MORGAN

$$\forall x \neg [F(x) \vee H(x)] \rightarrow G(x)$$
$$\neg F(a)$$
$$\frac{\neg H(a)}{G(a) \quad \therefore}$$

## 3.2 GENERALIZED CONTRAPOSITION

### 3.2.1 BASIC SCHEMA

$$\frac{\forall x F(x) \rightarrow \neg G(x)}{\forall x G(x) \rightarrow \neg F(x) \quad \therefore}$$

### 3.2.2 NEGATIVE VARIANT

$$\frac{\forall x F(x) \rightarrow G(x)}{\forall x \neg G(x) \rightarrow \neg F(x) \quad \therefore}$$

### 3.2.3 COMPLEX PREDICATES

$$\frac{\forall x [F(x) \wedge H(x)] \rightarrow \neg G(x)}{\forall x G(x) \rightarrow \neg [F(x) \wedge H(x)] \quad \therefore}$$

### 3.2.4 DE MORGAN

$$\frac{\forall x [F(x) \wedge H(x)] \rightarrow \neg G(x)}{\forall x G(x) \rightarrow \neg F(x) \vee \neg H(x) \quad \therefore}$$

## 3.3 HYPOTHETICAL SYLLOGISM 1

### 3.3.1 BASIC SCHEMA

$$\frac{\begin{array}{c} \forall x F(x) \rightarrow G(x) \\ \forall x G(x) \rightarrow H(x) \end{array}}{\forall x F(x) \rightarrow H(x) \quad \therefore}$$

### 3.3.2 NEGATIVE VARIANT

$$\frac{\begin{array}{c} \forall x F(x) \rightarrow \neg G(x) \\ \forall x \neg G(x) \rightarrow H(x) \end{array}}{\forall x F(x) \rightarrow H(x) \quad \therefore}$$

### 3.3.3 COMPLEX PREDICATES

$$\frac{\begin{array}{c} \forall x F(x) \rightarrow G(x) \\ \forall x F(x) \rightarrow I(x) \\ \forall x G(x) \wedge I(x) \rightarrow H(x) \end{array}}{\forall x F(x) \rightarrow H(x) \quad \therefore}$$

### 3.3.4 DE MORGAN

$$\frac{\begin{array}{c} \forall x \neg F(x) \wedge \neg I(x) \rightarrow G(x) \\ \forall x G(x) \rightarrow H(x) \end{array}}{\forall x \neg [F(x) \vee I(x)] \rightarrow H(x) \quad \therefore}$$

## 3.4 HYPOTHETICAL SYLLOGISM 2

### 3.4.1 BASIC SCHEMA

$$\frac{\begin{array}{c} \forall x F(x) \rightarrow G(x) \\ \forall x \neg H(x) \rightarrow \neg G(x) \end{array}}{\forall x F(x) \rightarrow H(x) \quad \therefore}$$

### 3.4.2 NEGATIVE VARIANT

$$\frac{\begin{array}{c} \forall x F(x) \rightarrow \neg G(x) \\ \forall x \neg H(x) \rightarrow G(x) \end{array}}{\forall x F(x) \rightarrow H(x) \quad \therefore}$$

### 3.4.3 COMPLEX PREDICATES

$$\forall x F(x) \to G(x) \lor I(x)$$
$$\forall x H(x) \to \neg[G(x) \lor I(x)]$$
$$\overline{\forall x F(x) \to H(x)} \quad \therefore$$

### 3.4.4 DE MORGAN

$$\forall x F(x) \to G(x) \lor I(x)$$
$$\forall x H(x) \to \neg G(x) \land \neg I(x)$$
$$\overline{\forall x F(x) \to H(x)} \quad \therefore$$

## 3.5 HYPOTHETICAL SYLLOGISM 3

### 3.5.1 BASIC SCHEMA

$$\forall x F(x) \to G(x)$$
$$\exists x H(x) \land \neg G(x)$$
$$\overline{\exists x H(x) \land \neg F(x)} \quad \therefore$$

### 3.5.2 NEGATIVE VARIANT

$$\forall x \neg F(x) \to G(x)$$
$$\exists x H(x) \land \neg G(x)$$
$$\overline{\exists x H(x) \land F(x)} \quad \therefore$$

### 3.5.3 COMPLEX PREDICATES

$$\forall x F(x) \to G(x)$$
$$\forall x F(x) \to I(x)$$
$$\exists x H(x) \land \neg[G(x) \land I(x)]$$
$$\overline{\exists x H(x) \land \neg F(x)} \quad \therefore$$

### 3.5.4 DE MORGAN

$$\forall x F(x) \to G(x)$$
$$\forall x F(x) \to I(x)$$
$$\exists x H(x) \land [\neg G(x) \lor \neg I(x)]$$
$$\overline{\exists x H(x) \land \neg F(x)} \quad \therefore$$

## 3.6 GENERALIZED MODUS TOLLENS

### 3.6.1 BASIC SCHEMA

$$\forall x F(x) \to G(x)$$
$$\neg G(a)$$
$$\overline{\neg F(a)} \quad \therefore$$

### 3.6.2 NEGATIVE VARIANT

$$\forall x F(x) \to \neg G(x)$$
$$G(a)$$
$$\overline{\neg F(a)} \quad \therefore$$

### 3.6.3 COMPLEX PREDICATES

$$\forall x F(x) \to G(x) \land H(x)$$
$$\neg G(a)$$
$$\overline{\neg F(a)} \quad \therefore$$

### 3.6.4 DE MORGAN

$$\forall x F(x) \to G(x) \land H(x)$$
$$\frac{\neg G(a) \lor \neg H(a)}{\neg F(a) \quad \therefore}$$

## 3.7 DISJUNCTIVE SYLLOGISM

### 3.7.1 BASIC SCHEMA

$$\forall x F(x) \to G(x) \lor H(x)$$
$$\frac{\forall x F(x) \to \neg G(x)}{\forall x F(x) \to H(x) \quad \therefore}$$

### 3.7.2 NEGATIVE VARIANT

$$\forall x F(x) \to G(x) \lor H(x)$$
$$\frac{\forall x G(x) \to \neg F(x)}{\forall x F(x) \to H(x) \quad \therefore}$$

### 3.7.3 COMPLEX PREDICATES

$$\forall x F(x) \to G(x) \lor H(x) \lor I(x)$$
$$\forall x F(x) \to \neg G(x)$$
$$\frac{\forall x F(x) \to \neg I(x)}{\forall x F(x) \to H(x) \quad \therefore}$$

### 3.7.4 DE MORGAN

$$\forall x F(x) \land I(x) \to G(x) \lor H(x)$$
$$\frac{\forall x G(x) \to \neg F(x) \lor \neg I(x)}{\forall x F(x) \land I(x) \to H(x) \quad \therefore}$$

## 3.8 GENERALIZED DILEMMA

### 3.8.1 BASIC SCHEMA

$$\forall x F(x) \to G(x) \lor H(x)$$
$$\forall x G(x) \to J(x)$$
$$\frac{\forall x H(x) \to J(x)}{\forall x F(x) \to J(x) \quad \therefore}$$

### 3.8.2 NEGATIVE VARIANT

$$\forall x F(x) \to G(x) \lor H(x)$$
$$\forall x J(x) \to \neg G(x)$$
$$\frac{\forall x J(x) \to \neg H(x)}{\forall x F(x) \to J(x) \quad \therefore}$$

### 3.8.3 COMPLEX PREDICATES

$$\forall x F(x) \to G(x) \lor H(x) \lor I(x)$$
$$\forall x J(x) \to \neg G(x)$$
$$\frac{\forall x J(x) \to \neg H(x)}{\forall x F(x) \to J(x) \lor I(x) \quad \therefore}$$

### 3.8.4 DE MORGAN

$$\forall x F(x) \to \neg[G(x) \land H(x)]$$
$$\forall x \neg G(x) \to J(x)$$
$$\frac{\forall x \neg H(x) \to J(x)}{\forall x F(x) \to J(x) \quad \therefore}$$

## 4   24 VALID TYPES OF SYLLOGISTIC REASONING

Each valid syllogism is given a name whose vowels indicate types of moods, e.g., 'CELARENT' indicates types of moods are 'E', 'A', 'E', respectively. 'A' for *universal affirmative*, all $X$ are $Y$, 'I' for *particular affirmative*, some $X$ are $Y$; 'E' for *universal negative*, no $X$ are $Y$, 'O' for *particular negative*, some $X$ are not $Y$. All 24 valid types of syllogistic reasoning is listed in Table 1.

## 5   SELF-SUPERVISED EULER NET

### 5.1   BENCHMARK DATASET OF EULER NET

Data sets of Euler Net are randomly created as follows. Let $R_{min}$ and $R_{max}$ be the pre-defined values denoting the minimum and the maximum lengths of radii. Let $\bigodot_A(O_A, R_A)$ represent circle A with the central point $O_A$ and the radius $R_A$. The Euler diagram where $\bigodot_A(O_A, R_A)$ is inside $\bigodot_B(O_B, R_B)$ is generated as follows: $R_A$ is randomly chosen between $R_{min}$ and $R_{max}$; $R_B$ is randomly chosen between $\lambda_1 R_{min}$ and $\lambda_2 R_A$ $(0 < \lambda_1 < \lambda_2 < 1)$; $O_A$ is fixed to $(R_{min} + R_{max}, R_{min} + R_{max})$; $O_B$ is set to $O_A + (\delta_x, \delta_y)$, in which $\delta_x$ and $\delta_y$ are randomly chosen between $(R_B - R_A)/2$ and $(R_A - R_B)/2$. Other Euler diagrams are generated in similar ways. In total, 96000 input-output pairs are created, which are separated into 80000 pairs for training, 8000 for validation, and 8000 for testing. Euler Net reached 99.8% accuracy in testing data.

### 5.2   AUTOMATIC SEARCHING FOR NEW TRAINING DATA

As illustrated in Figure 5 in our submitted paper, in each iteration, our self-supervised Euler Net firstly searches for new training data. The size of the newly created training data set in each iteration is set to 10,000. In the procedure of random search, the central point and the radius of a circle are totally random. We set two restrictions as follows: (1) circles are complete; (2) the minimum radius is set to 0.1. We allow all possible combinations between two circles. Following these criteria, we created two new testing data sets $\mathcal{D}_1$ and $\mathcal{D}_2$: $\mathcal{D}_1$ with one circle (half of input images in $\mathcal{D}_1$ have green circles, the rest of inputs are one red circle, one blue circle), $\mathcal{D}_2$ with two circles, and the size of $\mathcal{D}_2$ is 9 times larger than the size of $\mathcal{D}_1$. In each iteration, the newly trained Euler Net is tested with $\mathcal{D}_T = \mathcal{D}_1 \cup \mathcal{D}_2$. We set the maximum iteration number to 20.

### 5.3   EXPERIMENTAL RESULTS

We fed $\mathcal{D}_T$ to the original Euler Net, which achieved an accuracy of $56.0\%$ – this means that the original $99.8\%$ accuracy of Euler Net is only for **the benchmark datasets**. Through repeated training of newly created data sets, Euler Net improves its accuracy. It reaches a peak value of $97.8\%$ in accuracy after the $19^{th}$ iteration.

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

Table 1: List of all valid syllogisms, each is mapped to a qualitative spatial statement.

| Num | Name | Premise | Conclusion | Qualitative spatial relations statement |
|---|---|---|---|---|
| 1 | BARBARA | all $s$ are $m$, all $m$ are $p$ | all $s$ are $p$ | $\mathbf{P}(\mathcal{O}_s, \mathcal{O}_m) \wedge \mathbf{P}(\mathcal{O}_m, \mathcal{O}_p) \rightarrow \mathbf{P}(\mathcal{O}_s, \mathcal{O}_p)$ |
| 2 | BARBARI | all $s$ are $m$, all $m$ are $p$ | some $s$ are $p$ | $\mathbf{P}(\mathcal{O}_s, \mathcal{O}_m) \wedge \mathbf{P}(\mathcal{O}_m, \mathcal{O}_p) \rightarrow \neg\mathbf{D}(\mathcal{O}_s, \mathcal{O}_p)$ |
| 3 | CELARENT | no $m$ is $p$, all $s$ are $m$ | no $s$ is $p$ | $\mathbf{D}(\mathcal{O}_m, \mathcal{O}_p) \wedge \mathbf{P}(\mathcal{O}_s, \mathcal{O}_m) \rightarrow \mathbf{D}(\mathcal{O}_s, \mathcal{O}_p)$ |
| 4 | CESARE | no $p$ is $m$, all $s$ are $m$ | no $s$ is $p$ | $\mathbf{D}(\mathcal{O}_p, \mathcal{O}_m) \wedge \mathbf{P}(\mathcal{O}_s, \mathcal{O}_m) \rightarrow \mathbf{D}(\mathcal{O}_s, \mathcal{O}_p)$ |
| 5 | CALEMES | all $p$ are $m$, no $m$ is $s$ | no $s$ is $p$ | $\mathbf{P}(\mathcal{O}_p, \mathcal{O}_m) \wedge \mathbf{D}(\mathcal{O}_m, \mathcal{O}_s) \rightarrow \mathbf{D}(\mathcal{O}_s, \mathcal{O}_p)$ |
| 6 | CAMESTRES | all $p$ are $m$, no $s$ is $m$ | no $s$ is $p$ | $\mathbf{P}(\mathcal{O}_p, \mathcal{O}_m) \wedge \mathbf{D}(\mathcal{O}_s, \mathcal{O}_m) \rightarrow \mathbf{D}(\mathcal{O}_s, \mathcal{O}_p)$ |
| 7 | DARII | all $m$ are $p$, some $s$ are $m$ | some $s$ are $p$ | $\mathbf{P}(\mathcal{O}_m, \mathcal{O}_p) \wedge \neg\mathbf{D}(\mathcal{O}_s, \mathcal{O}_m) \rightarrow \neg\mathbf{D}(\mathcal{O}_s, \mathcal{O}_p)$ |
| 8 | DATISI | all $m$ are $p$, some $m$ are $s$ | some $s$ are $p$ | $\mathbf{P}(\mathcal{O}_m, \mathcal{O}_p) \wedge \neg\mathbf{D}(\mathcal{O}_m, \mathcal{O}_s) \rightarrow \neg\mathbf{D}(\mathcal{O}_s, \mathcal{O}_p)$ |
| 9 | DARAPTI | all $m$ are $s$, all $m$ are $p$ | some $s$ are $p$ | $\mathbf{P}(\mathcal{O}_m, \mathcal{O}_s) \wedge \mathbf{P}(\mathcal{O}_m, \mathcal{O}_p) \rightarrow \neg\mathbf{D}(\mathcal{O}_s, \mathcal{O}_p)$ |
| 10 | DISAMIS | some $m$ are $p$, all $m$ are $s$ | some $s$ are $p$ | $\neg\mathbf{D}(\mathcal{O}_m, \mathcal{O}_p) \wedge \mathbf{P}(\mathcal{O}_m, \mathcal{O}_s) \rightarrow \neg\mathbf{D}(\mathcal{O}_s, \mathcal{O}_p)$ |
| 11 | DIMATIS | some $p$ are $m$, all $m$ are $s$ | some $s$ are $p$ | $\neg\mathbf{D}(\mathcal{O}_p, \mathcal{O}_m) \wedge \mathbf{P}(\mathcal{O}_m, \mathcal{O}_s) \rightarrow \neg\mathbf{D}(\mathcal{O}_s, \mathcal{O}_p)$ |
| 12 | BAROCO | all $p$ is $m$, some $s$ are not $m$ | some $s$ are not $p$ | $\mathbf{P}(\mathcal{O}_p, \mathcal{O}_m) \wedge \neg\mathbf{P}(\mathcal{O}_s, \mathcal{O}_m) \rightarrow \neg\mathbf{P}(\mathcal{O}_s, \mathcal{O}_p)$ |
| 13 | CESARO | no $p$ is $m$, all $s$ are $m$ | some $s$ are not $p$ | $\mathbf{D}(\mathcal{O}_p, \mathcal{O}_m) \wedge \mathbf{P}(\mathcal{O}_s, \mathcal{O}_m) \rightarrow \neg\mathbf{P}(\mathcal{O}_s, \mathcal{O}_p)$ |
| 14 | CAMESTROS | all $s$ are $m$, no $m$ is $p$ | some $s$ are not $p$ | $\mathbf{P}(\mathcal{O}_s, \mathcal{O}_m) \wedge \mathbf{D}(\mathcal{O}_m, \mathcal{O}_p) \rightarrow \neg\mathbf{P}(\mathcal{O}_s, \mathcal{O}_p)$ |
| 15 | CELARONT | no $s$ is $m$, all $p$ are $m$ | some $s$ are not $p$ | $\mathbf{D}(\mathcal{O}_s, \mathcal{O}_m) \wedge \mathbf{P}(\mathcal{O}_p, \mathcal{O}_m) \rightarrow \neg\mathbf{P}(\mathcal{O}_s, \mathcal{O}_p)$ |
| 16 | CALEMOS | all $p$ are $m$, no $m$ is $s$ | some $s$ are not $p$ | $\mathbf{P}(\mathcal{O}_p, \mathcal{O}_m) \wedge \mathbf{D}(\mathcal{O}_m, \mathcal{O}_s) \rightarrow \neg\mathbf{P}(\mathcal{O}_s, \mathcal{O}_p)$ |
| 17 | BOCARDO | some $m$ are not $p$, all $m$ are $s$ | some $s$ are not $p$ | $\neg\mathbf{P}(\mathcal{O}_m, \mathcal{O}_p) \wedge \mathbf{P}(\mathcal{O}_m, \mathcal{O}_s) \rightarrow \neg\mathbf{P}(\mathcal{O}_s, \mathcal{O}_p)$ |
| 18 | BAMALIP | all $m$ are $s$, all $p$ are $m$ | some $s$ are $p$ | $\mathbf{P}(\mathcal{O}_m, \mathcal{O}_s) \wedge \mathbf{P}(\mathcal{O}_p, \mathcal{O}_m) \rightarrow \neg\mathbf{D}(\mathcal{O}_s, \mathcal{O}_p)$ |
| 19 | FERIO | some $s$ are $m$, no $m$ is $p$ | some $s$ are not $p$ | $\mathbf{P}(\mathcal{O}_s, \mathcal{O}_m) \wedge \mathbf{D}(\mathcal{O}_m, \mathcal{O}_p) \rightarrow \neg\mathbf{P}(\mathcal{O}_s, \mathcal{O}_p)$ |
| 20 | FESTINO | some $s$ are $m$, no $p$ is $m$ | some $s$ are not $p$ | $\mathbf{P}(\mathcal{O}_s, \mathcal{O}_m) \wedge \mathbf{D}(\mathcal{O}_p, \mathcal{O}_m) \rightarrow \neg\mathbf{P}(\mathcal{O}_s, \mathcal{O}_p)$ |
| 21 | FERISON | some $m$ are $s$, no $m$ is $p$ | some $s$ are not $p$ | $\mathbf{P}(\mathcal{O}_m, \mathcal{O}_s) \wedge \mathbf{D}(\mathcal{O}_m, \mathcal{O}_p) \rightarrow \neg\mathbf{P}(\mathcal{O}_s, \mathcal{O}_p)$ |
| 22 | FRESISON | some $m$ are $s$, no $p$ is $m$ | some $s$ are not $p$ | $\mathbf{P}(\mathcal{O}_m, \mathcal{O}_s) \wedge \mathbf{D}(\mathcal{O}_p, \mathcal{O}_m) \rightarrow \neg\mathbf{P}(\mathcal{O}_s, \mathcal{O}_p)$ |
| 23 | FELAPTON | all $m$ are $s$, no $m$ is $p$ | some $s$ are not $p$ | $\mathbf{P}(\mathcal{O}_m, \mathcal{O}_s) \wedge \mathbf{D}(\mathcal{O}_m, \mathcal{O}_p) \rightarrow \neg\mathbf{P}(\mathcal{O}_s, \mathcal{O}_p)$ |
| 24 | FESAPO | all $m$ are $s$, no $p$ is $m$ | some $s$ are not $p$ | $\mathbf{P}(\mathcal{O}_m, \mathcal{O}_s) \wedge \mathbf{D}(\mathcal{O}_p, \mathcal{O}_m) \rightarrow \neg\mathbf{P}(\mathcal{O}_s, \mathcal{O}_p)$ |