# OpenReview forum: "Closing the Gap between Neural Networks for Approximate and Rigorous Logical Reasoning"
_ICLR.cc/2025/Conference — Submitted to ICLR 2025_

### Official Review · Reviewer_rCFh · 2024-11-01

**Soundness:** 1
**Presentation:** 2
**Contribution:** 2
**Rating:** 3
**Confidence:** 3

**Summary:**

The authors study whether current neural networks can perform robust syllogistic reasoning via Euler diagrams, showing that they fail in very specific aspects, and conclude with arguments stating that neural networks need to go beyond vector embeddings to solve rigorous reasoning.

**Strengths:**

The paper is fairly well written, with some clear figures, especially in the revision. It presents multiple interesting ideas and experiments on syllogistic reasoning, a simple but easy-to-study problem.

**Weaknesses:**

I found it hard to follow what the contributions of this paper are. There are a few results that seem simple, arbitrary, poorly explained, and relevant only to a single network architecture. It is not clear to me what I should take home from these experiments.

The 'sketched proof' which is supposed to prove that transformers cannot do syllogistic reasoning also falls short: It assumes that they oversmooth, which only happens for transformers with many layers (the theoretical results are for the infinite-depth setting). If this happened consistently in practical transformer models, there is no chance LLMs could work as well as they do (as also Dovonon 2024 argues and shows, which is cited).

Together, this paper only provides meagre evidence for the infeasibility of syllogistic reasoning. Then the authors argue that different concept embeddings are needed, but do not compare (either theoretically or empirically) to the vector case, except for referring quickly to related work.

**Questions:**

- What is the motivation for specifically studying this Siamese Masked Autoencoder model? I suppose that this model does not use specific embeddings for each object (unlike models in object-centric learning, involving eg slot attention [1] or the method specific for this task as cited [2])
- Line 357: "We fed new randomly generated test data' How is this data different?
- Line 359: What's the motivation for Euler Net version 2? The description of this method is extremely difficult to follow and incomplete. How does a model 'generate' input images?
- 4.1, first paragraph. This lacks in details. Furthermore, it's well known that standard NNs are not adversarially robust. This connection is missing.
- 4.2: I did not understand the point of this experiment. Of course a model will not be able to say anything meaningful about incorrect input data that we never defined how to respond to, especially if it's not designed for out of distribution detection.
- Line 428: This blanket statement is highly overclaiming these results. This is about misspecification - not a lack of learning capability.
- 4.3: It is not clear to me how these combination tables are defined from a neural network point of view. Furthermore, this result again comes from the design of the neural network. If it's allowed to output multiple answers (for instance like an LLM would be able to), it may give all syllogistic conclusions.
- 479 "More powerful than vanilla RNN, LSTM": From a theoretical perspective, this is hard to claim. RNNs (with unbounded time) are Turing Complete [3]. Similar results exist for Transformers, but these require an infinite 'scratchpad / chain of thought' [4]. I suppose this 'powerful' refers to an empirical interpretation, but this should be clarified.
- Theorem 1 is unclear and informal, and does not properly state its assumptions. What is oversmoothing? Output embeddings? "will be points"? Of course output embeddings are points. What are the assumptions on the model architecture? A quick look at the proof did not help me understand these questions. This certainly doesn't constitute a 'rigorous proof" (Line 531)
- Similarly for Theorem 2, I have no idea what "If the output embeddings are not points" would mean.

[1] Locatello, Francesco, et al. "Object-centric learning with slot attention." Advances in neural information processing systems 33 (2020): 11525-11538.

[2] Wang, Duo, Mateja Jamnik, and Pietro Lio. "Abstract diagrammatic reasoning with multiplex graph networks." arXiv preprint arXiv:2006.11197 (2020).

[3] Nowak, Franz, et al. "On the representational capacity of recurrent neural language models." arXiv preprint arXiv:2310.12942 (2023).

[4] Lena Strobl, William Merrill, Gail Weiss, David Chiang, Dana Angluin; What Formal Languages Can Transformers Express? A Survey. Transactions of the Association for Computational Linguistics 2024; 12 543–561.

---

> ### Author Response · Authors · 2024-11-13
>
> The takeaway is as follows:
>
> Current deep-learning systems cannot and will not reach the rigour of logical reasoning, no matter what kinds of and how much training data we use. To achieve the rigour of logical reasoning, traditional neural networks shall do qualitative extensions, namely, to promote vector embedding to non-vector embedding.
>
> The ‘sketched proof’ does not prove transformers cannot do syllogistic reasoning.
>
> LLMs work very well, in terms of language communication, but this does not follow that they can reason well. see the reference below.
>
> Evelina Fedorenko, Steven T. Piantadosi, and Edward A. F. Gibson (2024). Language is primarily a tool for communication rather than thought. In Nature.
>
> This paper takes syllogistic reasoning as the micro-world of rationality and shows current deep-learning systems cannot and will not reach the rigour of syllogistic reasoning.
>
> Siamese architectures are used for object recognition and for syllogistic reasoning. In both cases, they achieve excellent results. However, The phenomena described in Section 4.1 raise the problem -- These single green circles are different from the standard inputs (two circles). Surprising is that the well-trained Euler-Net may automatically complete a single green circle into standard inputs. For object recognition, this is a great capability – it can recognise objects by observing its partial image (we do not say, partial images are out-of-distribution inputs). But, for reasoning, this capability shall not be allowed, because the neural networks shall not add new premises.
>
> Line 357: new randomly generated test data have different distributions from the training data.
>
> Line 359: The motivation is to let Euler Net improve its performance by itself. It is not difficult to create an image with two circles, given two centre points and two radii.
>
> The theorem is solidly proved using region-based spatial logic. The proof shall be independent of model architectures.

---

> ### Comment · Reviewer_rCFh · 2024-11-14
>
> It is well known that NNs/transformers fail to reason out of distribution, see eg [1, 2, 3]. I don't debate this. My comment is on how this paper attempts to show that it is impossible to learn syllogistic reasoning, which I believe is poorly explained, incomplete and lacks evidence.
>
> > These single green circles are different from the standard inputs (two circles).
>
> Yes, since they're OOD. It will act like the distribution it is trained on - this is not surprising at all.
>
> > Line 357: new randomly generated test data have different distributions from the training data.
>
> Of course I get that, but nowhere it is specified _how_ this distribution is different.
>
> > The theorem is solidly proved using region-based spatial logic. The proof shall be independent of model architectures.
>
> The proof really needs more than this. There are hidden assumptions on what these neural architectures are and that they oversmooth (which again is undefined).
>
> I increased my confidence to reflect the rebuttal.
>
> [1] Zhang, Honghua, et al. "On the paradox of learning to reason from data." Proceedings of the Thirty-Second International Joint Conference on Artificial Intelligence. 2023.
>
> [2] Berglund, Lukas, et al. "The Reversal Curse: LLMs trained on “A is B” fail to learn “B is A”." The Twelfth International Conference on Learning Representations.
>
> [3] Mirzadeh, Iman, et al. "Gsm-symbolic: Understanding the limitations of mathematical reasoning in large language models." arXiv preprint arXiv:2410.05229 (2024).

---

> > ### Author Response · Authors · 2024-11-14
> >
> > You totally misunderstand our argument. We do not argue it (NN) is impossible to learn syllogistic reasoning. What we argue is: it (NN) is impossible to reach the rigour of the symbolic level of syllogistic reasoning.
> >
> > The symbolic level rigour means: for any syllogistic reasoning (two premises and one conclusion), the NN shall determine whether the three statements are satisfiable or unsatisfiable.
> >
> > For a simple supervised NN, if it is confident for the reasoning “all Greeks are humans. All humans are mortal. Therefore, all Greeks are mortal”, it will not be confident for the weaker reasoning  “all Greeks are humans. All humans are mortal. Therefore, SOME Greeks are mortal”.  However, both are valid syllogistic reasoning.
> >
> > To deduce the velocity to escape the earth gravity is approx 11km pro second, we do not need to examine the mechanics of bikes or cars (why they cannot). However, with this theoretical result, we can design rockets to escape the earth gravity and fly to the moon.
> >
> > The proof is a complete logical deduction. We do not and shall not consider engineering aspects of existing neural architectures.   With this theoretical result, we conclude (after negating the theorem) that one condition for NN to avoid oversmoothing (which is well-defined in the literature and we also referenced) is to use non-vector embedding. This is a pre-condition to reach the rigour (symbolic level) of logical reasoning (this level is much higher than the level discussed in the papers you listed). Our conclusion is consistent with the other research that we have referenced.

---

> > > ### Comment · Reviewer_rCFh · 2024-11-14
> > >
> > > > it (NN) is impossible to reach the rigour of the symbolic level of syllogistic reasoning.
> > >
> > > Transformers and RNNs are sufficiently powerful to represent logical reasoning **for fixed sizes of inputs** (which as far as I can tell is the problem you're studying in your paper). I shared existing expressivity results in my review on these architectures that go far beyond the complexity of this problem. Whether they will actually learn this is something else - that depends on your data and training setup (as I shared in the references in the comment).
> > >
> > > > However, both are valid syllogistic reasoning.
> > >
> > > This is just how you setup your NNs and data. Natural language inference and models tackling this is the task of judging whether a reasoning step holds and allows for multiple reasoning steps to be valid.
> > >
> > > > We do not and shall not consider engineering aspects of existing neural architectures
> > >
> > > Up to you. The problem you study is finite and can be solved by much weaker architectures than ones that exhibit oversmoothing (infinite / deep transformers or GNNs). The condition (which again is not well-defined in the paper) of oversmoothing is thus much too strong to claim impossibility results.

---

> > > > ### Author Response · Authors · 2024-11-14
> > > >
> > > > Our argument tells as long as Transformers and RNNs use (1) vector embeddings for concepts and (2) composition tables for mapping premises and conclusions, they will not reach the rigorr of the symbolic level of syllogistic reasoning. This is independent of what training data you use – the example that we gave in the last response shows there are no consistent training data for logical conclusion and logical consistent conclusion.
> > > >
> > > > “existing expressivity results in my review on these architectures that go far beyond the complexity of this problem”.
> > > > Exactly, they are beyond the complexity of syllogistic reasoning. That is the reason we choose syllogistic reasoning to explore whether and how traditional NN (including Transformers and RNNs) can reach rigorous reasoning. If they cannot for simple syllogistic reasoning, as the micro-world for rational reasoning, they will not for other rational reasoning.
> > > >
> > > > Syllogistic reasoning is special – it dominated the research of logic for 2000 years, and the research of rationality in psychology for over 100 years (till today, it is still not completely solved). Solving neural syllogistic reasoning is the first step to solve more complex reasoning problems.
> > > >
> > > > > This is just how you setup your NNs and data. Natural language inference and models tackling this is the task of judging whether a reasoning step holds and allows for multiple reasoning steps to be valid.
> > > >
> > > > No. Syllogistic reasoning is atomic. We cannot break it into multiple steps, they cannot either. The current methods (using training data) cannot determine there is no model (unsatisfiability), and thus cannot separate satisfiable reasoning from valid reasoning and reach the rigor of symbolic level of logic reasoning.
> > > >
> > > > >. Up to you.
> > > >
> > > > No, it is up to whether we want to be alchemists or modern engineers.
> > > >
> > > > >The problem you study is finite and can be solved by much weaker architectures than ones that exhibit oversmoothing (infinite / deep transformers or GNNs). The condition (which again is not well-defined in the paper) of oversmoothing is thus much too strong to claim impossibility results.
> > > >
> > > > No. The problem is finite and appears simple, but cannot be solved by both weaker and complex architectures (without qualitative extensions). Oversoomthing is defined in line 502-503 and also in line 507 again -- outputs converge to the same feature embedding.
> > > >
> > > > No matter whether the condition of oversoomthing is strong or not, our proof (the rigorous logic deduction) guarantees the results.

---

> > > > > ### Author Response · Authors · 2024-11-15
> > > > >
> > > > > We hope you have read our arguments and explanation. At this stage, we win the debates. Do you have further questions or anything hard to accept?

---

> ### Comment · Reviewer_rCFh · 2024-11-15
>
> None of my concerns are properly addressed, so I'm keeping my score.
>
> To be clear, I don't disagree much with the _opinion_ on the feasibility of learning general reasoning. The point is that this is not a debate on this problem. I act as a reviewer of the paper, which I do not believe is in a state that is ready for publication, as I believe the evidence presented is not strong enough and is not clearly described.
>
> > No. The problem is finite and appears simple, but cannot be solved by both weaker and complex architectures (without qualitative extensions).
>
> With the right data, any finite problem can be solved by training NNs by just learning a direct input-output map.
>
> > Oversoomthing is defined in line 502-503 and also in line 507 again -- outputs converge to the same feature embedding.
>
> That's not a formal definition. What is the notion of convergence? What are the outputs? What is the process with which they converge? As far as I know, oversmoothing converges in the depth, meaning limited-depth models do not oversmooth. Limited depth models can solve any finite problem; hence they can solve the _finite_ problem of syllogistic reasoning.

---

> > ### Author Response · Authors · 2024-11-15
> >
> > >With the right data, any finite problem can be solved by training NNs by just learning a direct input-output map.
> > >Limited depth models can solve any finite problem; hence they can solve the finite problem of syllogistic reasoning.
> >
> > No. Here, we clearly argued that there is no right data to reach the rigour of symbolic level of syllogistic reasoning. We have explained this. It seems you neglected it. Again, taking two simplest syllogistic reasoning types:
> >
> > “all Greeks are humans. All humans are mortal. Therefore, ALL Greeks are mortal”
> >
> >  “all Greeks are humans. All humans are mortal. Therefore, SOME Greeks are mortal”
> >
> > What is the direct input-output map?
> >
> >
> > > That's not a formal definition.
> >
> > The formal definition of oversmoothing is abstracted into all output embedddings are coincided and described in the supplementary material.

---

> > ### Author Response · Authors · 2024-11-18
> >
> > > [1] Locatello, Francesco, et al. "Object-centric learning with slot attention." Advances in neural information processing systems 33 (2020): 11525-11538.
> >
> > Using Slot Attention mechanism, Euler Net will be more reliable to recognise the components in the input images and to separate intended inputs from unintended inputs. But, this will not change the content of the composition table (mapping intended inputs to outputs). In Figure 6, we show that composition tables can not cover all valid types of syllogistic reasoning.
> >
> > > [2] Wang, Duo, Mateja Jamnik, and Pietro Lio. "Abstract diagrammatic reasoning with multiplex graph networks." arXiv preprint arXiv:2006.11197 (2020).
> >
> > The Euler Net in this paper also used composition tables and covered 75% of valid types of syllogistic reasoning.
> >
> > > [3] Nowak, Franz, et al. "On the representational capacity of recurrent neural language models." arXiv preprint arXiv:2310.12942 (2023).
> >
> > > [4] Lena Strobl, William Merrill, Gail Weiss, David Chiang, Dana Angluin; What Formal Languages Can Transformers Express? A Survey. Transactions of the Association for Computational Linguistics 2024; 12 543–561.
> >
> > The two papers are about the linguistic expressive or representation powers of RNN and Transformers. Syllogistic statements are simple expressions for both. But, being able to determine whether statements are syllogistic statements does not follow being able to determine internal relations among these syllogistic statements.
> >
> > Roughly, given a finite set of words {Greek, human, mortal, X, Y, no, all}, NN can decide “all Greeks are human. no X are Y. all Greeks are mortal” are correct syllogistic statements at the level of syntax. They do not address the question whether the three statements form a valid reasoning.
> >
> > You raised a valuable issue – we will reference your listed papers and separate our work from them.

---

> > ### Author Response · Authors · 2024-11-19
> >
> > > That's not a formal definition. What is the notion of convergence? What are the outputs? What is the process with which they converge?
> >
> > Syllogistic reasoning in the real world will be involved with objects with so complex shapes and colours, that we need to have multi-layered Slot Attention Transformers to recognise them (each object embedding is a Gaussian distribution N(&mu;, &delta;))[1]. Then, we feed these object embeddings into a neural component to reason syllogistic relations, e.g. part-whole relations.
> >
> > Transformers suffer from oversmoothing when their depth increases. Oversmoothing means that outputs converge to the same feature embedding [2-5]. In our setting, oversmoothing means that all &mu; and all &delta; are the same. Consider the sphere O with centre &mu; and the radius &delta; as the representative of an object embedding. We formally define oversmoothing as all spheres are the same. In the setting of our syllogistic reasoning, we have ''for any O_i, if O_i is part of O_1, then O_i and O_1 are coincided.'' We prove that &delta; = 0, which is equivalent to that all spheres (also Gaussian distributions) are degraded into a point.
> >
> >
> > [1] Locatello, Francesco, et al. "Object-centric learning with slot attention." Advances in neural information processing systems 33 (2020): 11525-11538.
> >
> > [2] Namuk Park and Songkuk Kim. How do Vision Transformers Work? In ICLR 2022.
> >
> > [3] Pei hao Wang, Wen qing Zheng, Tian long Chen, and Zhang yang Wang. Anti-oversmoothing in deep vision transformers via the fourier domain analysis: From theory to practice. In ICLR 2022.
> >
> > [4] Xiao Jun Guo, Yi Fei Wang, Tian Qi Du, and Yi Sen Wang. Contranorm: A contrastive learning perspective on oversmoothing and beyond. In ICLR 2023
> >
> > [5] Gbètondji J-S Dovonon, Michael M. Bronstein, and Matt J. Kusner. Setting the record straight on transformer oversmoothing, 2024. https://arxiv.org/abs/2401.04301

---

> > > ### Author Response · Authors · 2024-11-21
> > >
> > > Dear Reviewer,
> > >
> > > We go through all your comments as follows (part 1).
> > >
> > > >There are a few results that seem simple, arbitrary, poorly explained, and relevant only to a single network architecture.
> > >
> > > From this single network architecture, we generalise the limitation of combination tables -- they can not cover all types of valid syllogistic reasoning. This method causes all network architectures, if they use composition tables, not to cover all types of valid syllogistic reasoning.
> > >
> > > > It is not clear to me what I should take home from these experiments.
> > > Here,  we show that existing neural network architectures cannot reach rigorous syllogistic reasoning.  This may surprise many researchers.
> > >
> > > > The 'sketched proof' which is supposed to prove that transformers cannot do syllogistic reasoning also falls short: It assumes that they oversmooth, which only happens for transformers with many layers (the theoretical results are for the infinite-depth setting). If this happened consistently in practical transformer models, there is no chance LLMs could work as well as they do (as also Dovonon 2024 argues and shows, which is cited).
> > >
> > > LLMs indeed perform well with syllogistic reasoning but have not reached the rigour of syllogistic reasoning [1,2].
> > >
> > > [1] Tiwalayo Eisape, MH Tessler, Ishita Dasgupta, Fei Sha, Sjoerd van Steenkiste, and Tal Linzen. A systematic comparison of syllogistic reasoning in humans and language models. In NAACL, 2024
> > >
> > > [2] Andrew K Lampinen, Ishita Dasgupta, Stephanie C Y Chan, Hannah R Sheahan, Antonia Creswell, Dharshan Kumaran, James L McClelland, and Felix Hill. Language models, like humans, show content effects on reasoning tasks. PNAS Nexus, 3(7), 2024
> > >
> > > > Then the authors argue that different concept embeddings are needed, but do not compare (either theoretically or empirically) to the vector case, except for referring quickly to related work.
> > >
> > > We do not compare, because this goes beyond the scope of this paper. Here, we mainly argue that traditional neural networks cannot reach the rigour of syllogistic reasoning (because most researchers may believe this task is easy and already solved by RNN or Transformers). Then, we theoretically prove a theorem that shows a necessary step to connect the recent work [1] that reaches the rigour of syllogistic reasoning.
> > >
> > > > What is the motivation for specifically studying this Siamese Masked Autoencoder model?
> > >
> > > We show that recovering the whole by observing its parts is desirable for object recognition. This is approximately simulated by the Siamese Masked Autoencoder model. The same architecture is used in Euler Net for syllogistic reasoning. Euler Net demonstrated the desirable ability in object recognition (observing a single green circle and recognising a regular input with two circles), but this ability is not desirable for reasoning.
> > >
> > > We try to convey the message that the object recognition component will cause problems for an end-to-end NN system for reasoning.
> > >
> > > > Line 357: "We fed new randomly generated test data' How is this data different?
> > >
> > > These new data were generated by randomly choosing the centres and the radii of two circles (as long as they are complete in the image). Two images in the original dataset of Euler Net were described in Section 5.1 in the supplementary material – line 389 – 395 – they were also randomly generated, but with different constraints.
> > >
> > > > Line 359: What's the motivation for Euler Net version 2? The description of this method is extremely difficult to follow and incomplete. How does a model 'generate' input images?
> > >
> > > The motivation of Euler Net version 2 is to explore the reasoning limit of Euler Net. The idea is to implement a wrapper system that can decide whether the output of Euler Net is correct, and if not, this wrapper system will create a new piece of training data for this error. The random input generator of the wrapper system chooses the centres and radii of two circles, and then it can compute the (expected) correct output of Euler Net. Using the set of automatically generated new training data, Euler Net is trained and evolves to Version 2.
> > >
> > > > 4.1, first paragraph. This lacks in details. Furthermore, it's well known that standard NNs are not adversarially robust. This connection is missing.
> > >
> > > We show that Euler Net demonstrated good capability in object recognition and this caused problems for reasoning. For reasoning, a single green circle is OOD, but for object recognition, it is not – it exists in the training data.

---

> > > > ### Author Response · Authors · 2024-11-21
> > > >
> > > > Dear Reviewer,
> > > >
> > > > We go through all your comments as follows (part 2).
> > > >
> > > > > 4.2: I did not understand the point of this experiment. Of course a model will not be able to say anything meaningful about incorrect input data that we never defined how to respond to, especially if it's not designed for out of distribution detection.
> > > >
> > > > Here, we show another limitation of supervised neural networks for reasoning – we cannot exhaust unintended input data, as intended input data can automatically create them. We can extend the set of intended data, but the extended data will create new unintended data.
> > > >
> > > > On the other hand, recent research shows the possibility of developing neural networks that can achieve the rigour of syllogistic reasoning without using training data [3].
> > > >
> > > > [3]  T. Dong, M. Jamnik, P. Lió (2024), Sphere Neural Network for Rational Reasoning. https://arxiv.org/abs/2403.15297.
> > > >
> > > > > Line 428: This blanket statement is highly overclaiming these results. This is about misspecification - not a lack of learning capability.
> > > >
> > > > Our experiment shows that we cannot have a complete specification. We may specify them into one of the classes or define them as invalid. Each will generate new under-specified outputs.
> > > >
> > > > > 4.3 It is not clear to me how these combination tables are defined from a neural network point of view. Furthermore, this result again comes from the design of the neural network. If it's allowed to output multiple answers (for instance like an LLM would be able to), it may give all syllogistic conclusions.
> > > >
> > > > If we allow NN to output multiple answers, NN will give each answer a probability. The sum of these probabilities will be less than or equal to 1. To reach the rigour of logical reasoning, NN should assign each answer with the probability 1.
> > > >
> > > > > Line 479. RNNs (with unbounded time) are Turing Complete [3]. Similar results exist for Transformers..  I suppose this 'powerful' refers to an empirical interpretation, but this should be clarified.
> > > >
> > > > RNNs’ Turing Complete (similar to Transformers) are about linguistic expressiveness, not about their reasoning ability. Given a finite set of words {Greek, human, mortal, X, Y, no, all}, RNN can decide “all Greeks are human. no X are Y. all Greeks are mortal” are correct syllogistic statements. They do not address the question whether the three statements form a valid reasoning.
> > > >
> > > > Yes. This ‘powerful’ is an empirical interpretation. We referenced a recent survey of Transformer’s applications in Line 484.
> > > >
> > > > > Theorem 1 is unclear and informal, and does not properly state its assumptions. What is oversmoothing? Output embeddings? "will be points"? Of course output embeddings are points.
> > > >
> > > > The outputs of neural networks are not necessarily points. They can be boxes [4], Cones [5], Spheres [3].
> > > >
> > > > Our proof only focuses on the output representation and abstracts out the neural architecture. So, the theorem applies to any neural architecture, including Transformers.
> > > >
> > > > >  Theorem 2, I have no idea what "If the output embeddings are not points" would mean.
> > > >
> > > > “the output embeddings are not points” means that neural networks optimise extended geometric objects, instead of points, for example, boxes [4], Cones [5], Spheres [3].
> > > >
> > > > [4] H. Ren, W. Hu, J. Leskovec. Query2box: Reasoning over Knowledge Graphs in Vector Space Using Box Embeddings. ICLR, 2020.
> > > >
> > > > [5] Zhang, Zhanqiu and Wang, Jie and Chen, Jiajun and Ji, Shuiwang and Wu, Feng. ConE: Cone embeddings for multi-hop reasoning over knowledge graphs. NeurIPS 2021.

---

> > > > > ### Comment · Reviewer_rCFh · 2024-11-24
> > > > >
> > > > > I thank the authors for the detailed responses. Unfortunately, the responses do not address my core issues with the paper.
> > > > >
> > > > > > This method causes all network architectures, if they use composition tables, not to cover all types of valid syllogistic reasoning.
> > > > >
> > > > > Sure, but a neural network can be a map with multiple outputs, which can cover all valid types of syllogistic reasoning.
> > > > >
> > > > > > They do not address the question whether the three statements form a valid reasoning.
> > > > >
> > > > > Formal languages can encode decision questions, such as "Statement A, Statement B, Statement C?" Which gets accepted if the ? is a yes or no.
> > > > >
> > > > > > Transformers suffer from oversmoothing when their depth increases.
> > > > >
> > > > > Thanks for formalising this statement. Indeed: This assumption of increased depth is vital. However, the authors do not prove that deep transformers are needed: Can the single-hop task presented in this paper be solved with a single transformer layer? Since it's a finite task, I would assume so.
> > > > >
> > > > > > LLMs indeed perform well with syllogistic reasoning but have not reached the rigour of syllogistic reasoning [1,2].
> > > > >
> > > > > Sure, but that's a different task than the one with circles studied in the paper, as it involves natural language. Furthermore, these papers do not show the impossibility of this task, which is what is argued in the paper under review.
> > > > >
> > > > > > These new data were generated by randomly choosing the centres and the radii of two circles (as long as they are complete in the image). Two images in the original dataset of Euler Net were described in Section 5.1 in the supplementary material – line 389 – 395 – they were also randomly generated, but with different constraints.
> > > > >
> > > > > The paper would be improved if this was specified more clearly in the paper, for instance in the Appendix.
> > > > >
> > > > > >  The idea is to implement a wrapper system that can decide whether the output of Euler Net is correct, and if not, this wrapper system will create a new piece of training data for this error.
> > > > >
> > > > > The description here still raises many questions on how this is actually implemented. Pseudocode of this system would help.
> > > > >
> > > > > > If we allow NN to output multiple answers, NN will give each answer a probability. The sum of these probabilities will be less than or equal to 1. To reach the rigour of logical reasoning, NN should assign each answer with the probability 1.
> > > > >
> > > > > A neural network can have multiple binary outputs (multilabel classification), each giving the probability of the conclusion being true. These answers need not be exclusive.
> > > > >
> > > > > > RNNs’ Turing Complete (similar to Transformers) are about linguistic expressiveness, not about their reasoning ability.
> > > > >
> > > > > This is incorrect. A Turing complete formalism can implement any computational reasoning task, in fact, Turing completeness is how computability is defined...
> > > > >
> > > > > Finally, in a comment to another reviewer, the authors shared:
> > > > >
> > > > > > Many researchers may assume they can, at least for syllogistic reasoning (you may see the comments of our last reviewer)
> > > > >
> > > > > I only ever referred to the **single-hop** "syllogistic reasoning with circles" task studied in the paper under review. In fact, I shared that I agree in general.
> > > > >
> > > > > > To be clear, I don't disagree much with the opinion on the feasibility of learning general reasoning.

---

> > > > > > ### Author Response · Authors · 2024-11-24
> > > > > >
> > > > > > Thank you very much for the critical and precious comments.
> > > > > >
> > > > > > > a neural network can be a map with multiple outputs, which can cover all valid types of syllogistic reasoning.
> > > > > > > A neural network can have multiple binary outputs (multilabel classification), each giving the probability of the conclusion being true. These answers need not be exclusive.
> > > > > >
> > > > > > Yes, we agree that an NN can have multiple outputs. After a softmax operation, each output will have a probability. This is consistent with our argument that they do not reach the rigour of logical reasoning.  In the case of two output classes, each will have 50%. That is the probability of tossing a coin.
> > > > > >
> > > > > > A new issue is that these multiple outputs cannot distinguish logical conclusion from logical consistency. If the inputs are “all Greeks are human. All humans are mortal”, the outputs are “all Greeks are mortal” and “some Greeks are mortal”. The NN cannot learn that “all Greeks are mortal” (logical conclusion) is stronger than “some Greeks are mortal” (logical consistency). This method follows that we need to teach the NN all logical consistencies.
> > > > > >
> > > > > > > Can the single-hop task presented in this paper be solved with a single transformer layer? Since it's a finite task, I would assume so.
> > > > > >
> > > > > > Following the above discussion, a single transformer layer with multiple outputs can solve syllogistic reasoning probabilistically.
> > > > > >
> > > > > > > Formal languages can encode decision questions, such as "Statement A, Statement B, Statement C?" Which gets accepted if the ? is a yes or no.
> > > > > > >  This is incorrect. A Turing complete formalism can implement any computational reasoning task, in fact, Turing completeness is how computability is defined…
> > > > > >
> > > > > > You are right. We agree that formal languages (in symbolic AI) can encode decision questions and a Turing complete formalism (in symbolic AI) can implement any computational reasoning task. We should have written clearly – here, we talk about RNNs that can be Turing Complete (given unbounded time). As described in the paper [1], they (given unbounded computation time) can simulate any deterministic probabilistic Turing machine. But, it is still
> > > > > > a non-deterministic automaton [1. pp. 7017], so,  it will not reach the rigour of any logical reasoning. We will include this topic and discussion into the paper.
> > > > > >
> > > > > > [1] Nowak, Franz, et al. "On the representational capacity of recurrent neural language models." arXiv preprint arXiv:2310.12942 (2023).
> > > > > >
> > > > > > > these papers do not show the impossibility of this task, which is what is argued in the paper under review.
> > > > > >
> > > > > > Yes, these papers do not explicitly show the impossibility of LLMs to reach the rigour of syllogistic reasoning. But, these papers point out LLMs learn human errors in the training data. This implicitly follows that LLMs cannot reach the rigour of syllogistic reasoning.
> > > > > >
> > > > > > > The paper would be improved if this was specified more clearly in the paper, for instance in the Appendix.
> > > > > >
> > > > > > We will elaborate and move this part from the current supplementary material into the Appendix.
> > > > > >
> > > > > > > The description here still raises many questions on how this is actually implemented. Pseudocode of this system would help.
> > > > > >
> > > > > > We submitted codes (in pytorch) along with the paper.
> > > > > >
> > > > > > Here, we write some pseudocodes.
> > > > > >
> > > > > > ---
> > > > > > generate_one_input (rel, colour1, colour2)
> > > > > >
> > > > > > ```
> > > > > > circle1  ← randomly generate circle 1 with a 2-D point c1 as the centre and  a radius r1
> > > > > > circle2  ← randomly generate circle 2, with the centre c2 and radius r2, satisfying the set-thoretic relation rel with the first circle
> > > > > > image ← draw circle 1 (in colour 1)  and circle 2 (in colour 2)
> > > > > > return circle1(c1,r1), circle2(c2,r2), image
> > > > > > ```
> > > > > >
> > > > > > ---
> > > > > > generate_one_new_training_data (EulerNet, rel1, rel2)
> > > > > > ```
> > > > > > circle1(c1,r1), circle2(c2,r2), image1 ← generate_one_input (rel1, red, green)
> > > > > > circle1(c3,r3), circle2(c4,r4), image2 ← generate_one_input (rel2, green, blue)
> > > > > > expected_output ← get_set_theoretic_relation_between(circle1, circle4)
> > > > > > output_of_EN ← EulerNet(image1, image2)
> > > > > > if   output_of_EN is not equal with expected_output:
> > > > > > 	return ((image1, image2), expected_output)
> > > > > > else:
> > > > > > 	return empty
> > > > > > ```
> > > > > >
> > > > > > ---
> > > > > > collect_new_training_data(EulerNet, M)
> > > > > > ```
> > > > > > count = 0
> > > > > > training_data = []
> > > > > > for each  rel1 in the four set-theoretic relations:
> > > > > > 	for each  rel2 in the four set-theoretic relations:
> > > > > > 		new_data ← generate_one_new_training_data (EulerNet, rel1, rel2)
> > > > > > 		if new_data is not empty:
> > > > > > 			count += 1
> > > > > > 			training_data.append(new_data)
> > > > > > 		if count == M:
> > > > > > 			return  training_data
> > > > > > ```
> > > > > >
> > > > > > ---
> > > > > > automatic improvement of EulerNet for N times.
> > > > > > ```
> > > > > > version = 0
> > > > > > EN_0 ← EulerNet
> > > > > > while version < N:
> > > > > > 	training_data ← collect_new_training_data(EN_version, M)
> > > > > > 	EN_new ← train EN_version using  training_data
> > > > > > 	version += 1
> > > > > > 	EN_version  = EN_new
> > > > > > return EN_version
> > > > > > ```

---

> > > > > > > ### Author Response · Authors · 2024-11-25
> > > > > > >
> > > > > > > > I only ever referred to the single-hop "syllogistic reasoning with circles" task studied in the paper under review.
> > > > > > > > A neural network can have multiple binary outputs (multilabel classification), each giving the probability of the conclusion being true. These answers need not be exclusive.
> > > > > > >
> > > > > > > If we use multiple output labels for the valid reasoning “all W are U. some V are W. therefore, some V are U”, in the case of "syllogistic reasoning with circles" task studied in the paper (Figure 6), we will map “some V are W” into three kinds of circle relations: (1) V circle inside W circle, (2) V circle partially overlaps with W circle, (3) V circle contains W circle, and map the conclusion (*) “some V are U” to the vector [1,1,1,0]. This means that “circle W is inside circle U” and (3) “V circle contains W circle” will also map to [1,1,1,0]. This output is inconsistent with the logical conclusion and the logical consistency of the two premises (“circle W is inside circle U” and “V circle contains W circle”). So, These answers need to be exclusive. That is another reason multiple labels will not work (achieve rigorous reasoning).
> > > > > > >
> > > > > > > > However, the authors do not prove that deep transformers are needed: Can the single-hop task presented in this paper be solved with a single transformer layer? Since it's a finite task, I would assume so.
> > > > > > >
> > > > > > > From line 486 to 493, we show the query-key-value table is a composition table automatically learned by a single transformer layer. If it is a vision transformer, the table can not enumerate all valid syllogistic reasoning types. If it is a transformer for symbolic inputs (sentences), it will do probabilistically and cannot promise to work correctly for out-of-distribution words. We will explicitly state this in the paper.
> > > > > > >
> > > > > > > If anything in the explanations is still unclear, please let us know. Thank you.

---

> > > > > > > > ### Author Response · Authors · 2024-11-28
> > > > > > > >
> > > > > > > > Dear Reviewer rCFh,
> > > > > > > >
> > > > > > > > Thank you again for your constructive feedback. We almost rewrite the whole text and upload a new version, hoping all your concerns are addressed.

---

> > > > > > > > > ### Author Response · Authors · 2024-11-30
> > > > > > > > >
> > > > > > > > > Dear Reviewer,
> > > > > > > > >
> > > > > > > > > Your continued feedback will be highly appreciated.

---

> ### Comment · Reviewer_rCFh · 2024-12-03
>
> I thank the authors for their extensive revision.
>
> On a brief look, this revision addresses some of my concerns with the paper. Still, since this is a large change to the paper ("almost rewrote the entire text"), and it is not clear what changes are made in this revision. Therefore, I suggest the paper goes through another full round of reviews, so multiple people can have a thorough look at it. I updated my score to reject.
>
> PS: I regret choosing the "strong reject" option initially. The number "1" next to it is unnecessarily harsh. My apologies for this.

---

> > ### Author Response · Authors · 2024-12-03
> >
> > We sincerely thank you for reading the revision and for raising the score. We made these changes by following your constructive comments and those of other reviewers (though it is an extensive revision). The fact that supervised deep learning cannot reach rigorous syllogistic reasoning is a bit harsh (also to us). However, knowing this fact on time and the right way to pursue neural reasoning will greatly help others and may save many social resources. If possible, please let us know what concerns have, and have not, been addressed in the revised version. Thanks again.

---

### Official Review · Reviewer_g989 · 2024-11-02

**Soundness:** 2
**Presentation:** 2
**Contribution:** 2
**Rating:** 5
**Confidence:** 4

**Summary:**

The paper discusses the "dual-process" theory of mind, highlighting the distinction between fast, intuitive thinking  and slower, more deliberate thinking. It conclude that LLMs and
Foundation Models built upon Transformers cannot reach the rigour of syllogistic
reasoning.
The article proposes a method of transforming syllogistic relationships into "part-whole relationships" and suggests using non-vector embeddings instead of traditional vector embeddings to avoid the problem of "oversmoothing." Oversmoothing can cause the outputs of neural networks to converge to similar embeddings, thereby affecting the accuracy of reasoning.

**Strengths:**

This paper attempts to analyze and study the reasoning capabilities of transformers, which is of great value. Additionally, the methods proposed in this paper possess certain innovative and theoretical significance.

**Weaknesses:**

1. This work lacks experimental validation and seems to be not fully complete.

2. The article is not clearly written. The abstract and introduction are somewhat verbose, and the key innovations and objectives are not clearly defined.

**Questions:**

In fact, enhancing the inference capabilities of neural networks is a very challenging task. Will merely changing traditional vector embeddings yield significant improvements, or can it lead to substantial advancements?

**Details Of Ethics Concerns:**

No Ethics Concerns.

---

> ### Author Response · Authors · 2024-11-12
>
> Promoting traditional vector embeddings into manifold embedding is the first step. The second step is to introduce the method of reasoning as model construction, see, Sphere Neural-Networks for Rational Reasoning https://arxiv.org/abs/2403.15297
>
> Here, we show the limitations of (1) the vector representation, and (2) the method of reasoning through combination tables. Both prevent neural networks from achieving rigorous reasoning, which goes beyond the statistic metrics -- more data experiments will not help.  Three statements being unsatisfiable (contradictory) is a topic of possibility, not probability -- no training data for deciding unsatisfiability.

---

> ### Author Response · Authors · 2024-11-22
>
> Thanks for the suggestion.
>
> We conducted a validation experiment to show that using combination table Euler Net (EN) cannot cover all valid types of syllogistic reasoning, and will add it to the supplementary material and publish the new dataset.
>
> We created a new dataset that covers all 24 valid types of syllogistic reasoning, to test a well-trained Euler Net (99.8% accuracy on the benchmark dataset).
>
> This dataset is created as follows: We group 24 _valid: syllogism types into 14 groups, as _no x are y_ has the same meaning with  _ no y are x_; and _some x are y_ has the same meaning with _some y are x_. For each group, we created 500 test cases by extracting hypernym relations from WordNet-3.0, each test case consisting of one true conclusion and one false conclusion, totalling 14000 syllogism reasoning tasks.
> In the hypernym structure,  _elementary\_particle.n.01_ is a descendent of _natural\_object.n.01_ and _artifact.n.01_ is not a descendent of _natural\_object.n.01_. So, we create the true syllogistic reasoning as: If _all elementary\_particle.n.01 are natural\_object.n.01_, _no artifact.n.01 are natural\_object.n.01_, then _all elementary\_particle.n.01 are not artifact.n.01_. The false syllogistic reasoning will be : If _all elementary\_particle.n.01 are natural\_object.n.01_, _no artifact.n.01 are natural\_object.n.01_, then _ some elementary\_particle.n.01 are artifact.n.01_.
>
> \begin{array}{|l|c|l|c|l|c|}
>  \\hline Valid\ Type & Accuracy & Valid\ Type& Accuracy& Valid\ Type & Accuracy \\\\\\hline
>   BARBARA  & 100\\% & BARBARI& 50\\% &BAROCO&66.7\\%  \\\\\\hline
>   BAMALIP & 50\\% & BOCARDO& 75\\% &CALEMES  &100\\%  \\\\\\hline
>  CAMESTROS  & 50\\% & CELARENT& 100\\% &CESARO &50\\%  \\\\\\hline
> CALEMO  & 50\\% & CESARE& 100\\% & CELARONT&50\\%  \\\\\\hline
>   DARAPTI   & 100\\% & DARII& 75\\% &DISAMIS &75\\%  \\\\\\hline
> FESAPO  & 100\\% & DATISI& 75\\% &DIMATIS&75\\%  \\\\\\hline
> FELAPTON & 100\\% &FERIO& 83.3\\% &FERISON &83.3\\%  \\\\\\hline
> CAMESTRES & 100\\% &FRESISON& 83.3\\% &FESTINO&83.3\\%  \\\\\\hline
> Overall&76\\% &&&  \\\\\\hline
> \end{array}

---

> > ### Author Response · Authors · 2024-11-30
> >
> > Dear Reviewer,
> >
> > thanks again for giving us feedbacks. We have uploaded a new version of the paper with more experiments and examples. We would very much appreciate your continued feedback.

---

### Official Review · Reviewer_83YV · 2024-11-03

**Soundness:** 2
**Presentation:** 3
**Contribution:** 1
**Rating:** 5
**Confidence:** 3

**Summary:**

The authors highlight the limitations of neural networks, including large language models (LLMs), in achieving rigorous syllogistic reasoning, which is essential for logic and human rationality. They argue that these networks should avoid combination tables and instead use non-vector embeddings to prevent oversmoothing. The paper reviews the Siamese Masked Autoencoder and presents experiments demonstrating that models relying on combination tables cannot attain 100% accuracy in syllogistic tasks. However, using non-vector embeddings as computational building blocks can help neural networks avoid oversmoothing. This work aims to bridge the gap between neural networks for approximate and rigorous logical reasoning.

**Strengths:**

- The authors substantiate their claims with experimental results, showcasing the shortcomings of existing models, such as the Siamese Masked Autoencoder, in achieving high accuracy in syllogistic reasoning.
- The paper opens avenues for further exploration, encouraging researchers to develop architectures that can effectively address rigorous reasoning tasks.

**Weaknesses:**

The authors claim three main contributions, and there are corresponding weaknesses for each:

   - **Contribution 1:** The authors conduct an experiment in Section 4. However, the experiments in Sections 4.1 and 4.2 appear to primarily test neural models' performance on out-of-distribution inputs. The poor performance of neural models on out-of-distribution inputs is already well-documented, which limits the novelty of this contribution.

   - **Contribution 2:** The use of combination tables is discussed in Section 4.3, but this section is confusing. For example, the authors state that the combination table only generates the conclusion "all V are U" is not enough, since it misses the conclusion “some V are U.” However, the statement "all V are U" clearly describes a part-whole relationship, and "some V are U" can be derived from "all V are U." The authors did not explain why this senario is worse.

   - **Contribution 3:** The authors discuss this in Section 5 (lines 502-519), but the proof is unclear. For example, it's unclear how the two theorems prove "using non-vector feature embedding to avoid oversmoothing". Additionally there lacks empirical studies to support it.

**Questions:**

1. Are the phenomena described in Section 4.1 distinct from typical out-of-distribution scenarios?

2. In Section 5 (lines 502-519), what is the relationship between using (non-)vector feature embeddings and output embeddings being points?

3. Given that symbolic approaches are effective for syllogistic reasoning, why is it necessary for neural models to also support rigorous reasoning? In Section 2.1 (line 181), the authors argue that "symbolic approaches neither explain how symbols emerge from our neural minds nor capture the ways humans reason in daily life." Can neural models genuinely achieve these objectives?

---

> ### Author Response · Authors · 2024-11-12
>
> Yes. The phenomena described in Section 4.1 are entirely different from the out-of-distribution scenarios. These single green circles are different from the standard inputs (two circles); in this way, they are out-of-distribution and will perform incorrectly, as we expect. Surprising is that the well-trained Euler-Net may automatically complete a single green circle into standard inputs. For object recognition, this is a great capability – it can recognise objects by observing its partial image (we do not say, partial images are out-of-distribution inputs). But, for reasoning, this capability shall not be allowed, because the neural networks shall not add new premises.
>
> If using (non-)vector or vector feature embeddings and the output embeddings oversmooth, then the converged output embedding must be a single vector feature embedding (a point). Or, put it this way: if feature embeddings are spheres with radii >=0, and output embeddings oversmooth, then their radii = 0. This means if we restrict radii > 0, oversmoothing will not happen.
>
> After researchers promote vector embeddings into spheres and introduce the method of reasoning using model construction, neural models achieve rigorous syllogistic reasoning without training data, see, Sphere Neural-Networks for Rational Reasoning https://arxiv.org/abs/2403.15297

---

> > ### Comment · Reviewer_83YV · 2024-11-19
> >
> > For each part of your answer:
> > - I am afraid your reply isn't convincing enough.
> >   As you said, these single green circles are out-of-distribution. When processing OOD data, the model's behavior can be erratic, producing unexpected outputs. Therefore, the model can behave incorrectly on an image with a single green circle.
> >   In line 371, you said, "there were input images with only one green circle........" It is important to clarify if this behavior is consistent. If not, and similar OOD images lead to different or inconsistent errors, this simply reflects the model's unpredictable behavior which is typical of OOD scenarios.
> >
> > - I would appreciate a more rigorous explanation, particularly with relevant citations or explanations for any key terms introduced.
> >
> > - I don't think your answer addresses my question. Do you mean no requirement for training data is an advantage of neural models?

---

> > > ### Author Response · Authors · 2024-11-19
> > >
> > > Thank you very much for the constructive suggestions.
> > >
> > > > single green circles are out-of-distribution. … When processing OOD data, the model's behavior can be erratic, producing unexpected outputs. "there were input images with only one green circle.....…"
> > >
> > > If the regular inputs are images with two circles, images with a single
> > > green circle will be out-of-distribution inputs and the model may have
> > > erratic behaviour (we all know this.)
> > >
> > > Here, we used the method of maximisation activation (line 369) [1] and found
> > > that given particular single green circle inputs, Euler Net can have regular
> > > outputs, e.g., the red circle is inside the blue circle. The only possible
> > > explanation is that the latent vector embeddings of two single green circles
> > > are similar to one pair of regular inputs (one image with a red circle and a
> > > green circle, the other image with a green circle and a blue circle), and
> > > Euler Net used this pair of regular inputs to produce a regular output.
> > > This means that the well-trained Euler Net can automatically recognise the
> > > whole from the partial inputs. Recent research on object recognition
> > > supported this capability of Siamese network architecture (Section 3.1).
> > >
> > >
> > > This special out-of-distribution input let Euler Net demonstrate a desirable
> > > capability for object recognition and produce an unintended output for
> > > logical reasoning.
> > >
> > > > Do you mean no requirement for training data is an advantage of neural models?
> > >
> > > Yes. no requirement for training data is an advantage for developing
> > > computational models for reason [2-7], because (human) reasoning is a
> > > process of model construction and inspection. A recent research shows that
> > > the basis of (human) reasoning is about possibility [8], while training data
> > > is about probability that needs the stable-world assumption -- the training
> > > data and the testing data share the same distribution [10-11].  To develop
> > > neural models for high-level cognition, we shall go beyond the scope of this
> > > statistic paradigm [10]. If not, here, we show the impossibility to reach
> > > the rigour of syllogistic reasoning.
> > >
> > >
> > > [1] WojciechSamek, GrégoireMontavon, AndreaVedaldi, LarsKaiHansen, and
> > > Klaus-RobertMüller (eds.). ExplainableAI: Interpreting, Explaining and
> > > Visualizing Deep Learning, volume 11700 of Lecture Notes in Computer
> > > Science. 2019.
> > >
> > > [2] N. Johnson-Laird, R. M. J. Byrne, Deduction, Lawrence Erlbaum
> > > Associates, Inc., 1991.
> > >
> > > [3] M. Knauff, T. Fangmeier, C. C. Ruff, P. N. Johnson-Laird, Reasoning,
> > > models, and images: behavioral measures and cortical activity, Journal of
> > > Cognitive Neuroscience 15 (4) (2003) 559–573
> > >
> > > [4] G. Goodwin, P. Johnson-Laird, Reasoning about relations., Psychological
> > > review 112 (2005) 468–93.
> > >
> > > [5] M. Knauff, A neuro-cognitive theory of deductive relational reasoning
> > > with mental models and visual images, Spatial Cognition & Computation 9 (2)
> > > (2009) 109–137.
> > >
> > > [7] M. Ragni, M. Knauff, A theory and a computational model of spatial
> > > reasoning with preferred mental models, Psychological review 120 (2013)
> > > 561–588.
> > >
> > > [8] Johnson-Laird, P.N., Byrne, R.M.J. & Khemlani, S.S. Models of
> > > Possibilities Instead of Logic as the Basis of Human Reasoning. Minds &
> > > Machines 34, 19 (2024).
> > >
> > > [9] Mercier, D. Sperber, The Enigma of Reason, Penguin, 2018.
> > >
> > > [10] Anirudh Goyal and Y. Bengio. Inductive biases for deep learning of
> > > higher-level cognition. Proceedings of the Royal Society A: Mathematical,
> > > Physical and Engineering Sciences, 478, 10 2022.
> > >
> > > [11] Gerd Gigerenzer. How to Stay Smart in a Smart World: Why Human
> > > Intelligence Still Beats Algorithms. The MIT Press, 2022.

---

> > > ### Author Response · Authors · 2024-11-30
> > >
> > > Dear Reviewer,
> > >
> > > thanks again for giving us feedbacks. We have uploaded a new version of the paper and hope all your concerns are addressed. We would very much appreciate your continued feedback.

---

### Official Review · Reviewer_pFMU · 2024-11-03

**Soundness:** 2
**Presentation:** 2
**Contribution:** 3
**Rating:** 5
**Confidence:** 2

**Summary:**

This paper proposes a task that converts syllogism into subset relations and then generates an image dataset that visualizes the subset relations and evaluates neural networks. The authors show in their experiments that although Euler Networks can learn part-whole relations between two entities, it cannot learn complex combinations of these relations, resulting in a lack of validity in the equivalent syllogism reasoning. Furthermore, the authors hypothesized that NNs should use one-hot representation to acquire the rigorous reasoning ability.

**Strengths:**

- The paper presents an important question that the community really cares about.
- The author shows the equivalence between syllogism reasoning and part-whole relations, and converted reasoning task into a visual prediction problem, which is interesting to me.

**Weaknesses:**

- This paper still lacks enough experiments to support the authors' claims. Why would a one-hot representation save neural nets in reasoning soundness issues?
- The presentation of this paper could be further improved. The structure of it now looks more like a technical report. It lacks of figures and charts to present the experimental results.
- The discuss is high-level, while the technical detail or insufficiency of the compared methods are not discussed enough.

**Questions:**

Please see above.

---

> ### Author Response · Authors · 2024-11-12
>
> We are not sure whether we correctly understand your question.  One-hot representation reduces the amount of training data, compared with using image representation.

---

> ### Author Response · Authors · 2024-11-22
>
> Thanks for the suggestion.
>
> We conducted an additional experiment to show that using combination table Euler Net (EN) cannot cover all valid types of syllogistic reasoning, and will add it to the supplementary material and publish the new dataset.
>
> We created a new dataset that covers all 24 valid types of syllogistic reasoning, to test the performance of a well-trained Euler Net (99.8% accuracy on the benchmark dataset).
>
> This dataset is created as follows: We group 24 _valid_ syllogism types into 14 groups, as _no x are y_ has the same meaning with  _no y are x_; and _some x are y_ has the same meaning with _some y are x_. For each group, we created 500 test cases by extracting hypernym relations from WordNet-3.0, each test case consisting of one true conclusion and one false conclusion, totalling 14000 syllogism reasoning tasks.
> In the hypernym structure,  _elementary\_particle.n.01_ is a descendent of _natural\_object.n.01_ and _artifact.n.01_ is not a descendent of _natural\_object.n.01_. So, we create the true syllogistic reasoning as: If _all elementary\_particle.n.01 are natural\_object.n.01_, _no artifact.n.01 are natural\_object.n.01_, then _all elementary\_particle.n.01 are not artifact.n.01_. The false syllogistic reasoning will be : If _all elementary\_particle.n.01 are natural\_object.n.01_, _no artifact.n.01 are natural\_object.n.01_, then _some elementary\_particle.n.01 are artifact.n.01_.
>
> We use the pre-processing tool of EN to transform premises into coloured circles, and conclusions into vectors, respectively, and fed to EN. For 8 syllogistic structures, EN reaches 100\% accuracy, namely, BARBARA, CELARENT, CESARE, DARAPTI, CALEMES, CAMESTRES, FELAPTON, and FESAPO. Accuracies of the rest 16 types range from $50\\%$ to $83.3\\%$. The overall accuracy is 76\\%.
> \begin{array}{|l|c|l|c|l|c|}
>  \\hline Valid\ Type & Accuracy & Valid\ Type& Accuracy& Valid\ Type & Accuracy \\\\\\hline
>   BARBARA  & 100\\% & BARBARI& 50\\% &BAROCO&66.7\\%  \\\\\\hline
>   BAMALIP & 50\\% & BOCARDO& 75\\% &CALEMES  &100\\%  \\\\\\hline
>  CAMESTROS  & 50\\% & CELARENT& 100\\% &CESARO &50\\%  \\\\\\hline
> CALEMO  & 50\\% & CESARE& 100\\% & CELARONT&50\\%  \\\\\\hline
>   DARAPTI   & 100\\% & DARII& 75\\% &DISAMIS &75\\%  \\\\\\hline
> FESAPO  & 100\\% & DATISI& 75\\% &DIMATIS&75\\%  \\\\\\hline
> FELAPTON & 100\\% &FERIO& 83.3\\% &FERISON &83.3\\%  \\\\\\hline
> CAMESTRES & 100\\% &FRESISON& 83.3\\% &FESTINO&83.3\\%  \\\\\\hline
> Overall&76\\% &&&  \\\\\\hline
> \end{array}
>
> All these valid types are explained in the supplementary material.

---

> ### Comment · Reviewer_pFMU · 2024-11-24
>
> > One-hot representation reduces the amount of training data, compared with using image representation.
>
> Thank you for your answer, but this has nothing to do with logic soundness or validity. I don't think neural networks or statistical learning in general could guarantee soundness in logical reasoning, they are just probabilistic approximately correct.
>
> > We conducted an additional experiment to show that using combination table Euler Net (EN) cannot cover all valid types of syllogistic reasoning, and will add it to the supplementary material and publish the new dataset.
>
> Thank you for providing the detailed results, it improves the readability of your paper.
>
> But I think that Euler Network is a very special case in machine learning models. Your claim that it can learn one-hop part-of relational reasoning soundly (and in image representation) still lacks a theoretical guarantee.

---

> > ### Author Response · Authors · 2024-11-24
> >
> > > Thank you for your answer, but this has nothing to do with logic soundness or validity. I don't think neural networks or statistical learning in general could guarantee soundness in logical reasoning, they are just probabilistic approximately correct.
> >
> > Thank you for your valuable feedback. We also do not think supervised neural networks or statistical learning can guarantee soundness in logical reasoning. The aim of this paper is to argue why they cannot.
> > Many researchers may assume they can, at least for syllogistic reasoning (you may see the comments of our last reviewer) ,  in part because of the simple forms of syllogism and the Turing Complete of RNNs (with unbounded time).
> >
> > > But I think that Euler Network is a very special case in machine learning models. Your claim that it can learn one-hop part-of relational reasoning soundly (and in image representation) still lacks a theoretical guarantee.
> >
> > Euler Network is a special neural model in the literature designed particularly for syllogistic reasoning. It achieved 99.8% accuracy in the benchmark dataset. This is a statistical result -- If we create a new testing dataset having a different distribution from the benchmark, the performance of Euler Network will drop to 56% (line 358). We do not claim that Euler Net can learn one-hop part-of relational reasoning soundly (and in image representation). We agree that it can do this empirically and argue it cannot achieve rigorous syllogistic reasoning.
> >
> > You probably mean, “Euler Network is a very special case, so we cannot draw a strong conclusion that supervised neural network cannot reach rigorous syllogistic reasoning”. The strategy of our argument is as follows: we analyse the reasons from this special case. One reason is the use of composition tables to establish premise-conclusion relations. The family of neural networks that use this method will not reach rigorous syllogistic reasoning.
> >
> > > Why would a one-hot representation save neural nets in reasoning soundness issues?
> >
> > Let us answer your original question, by comparing two formats of training data: one is ((input image 1, input image 2), output one-hot representation), the other is one is ((input image 1, input image 2), output image), one output one-hot representation will correspond to many possible output images. If one one-hot representation corresponds to 1000 images, the amount of the training data in the second format will be 1000 times more. An additional question is how to interpret this output image – We need an extra network that maps an output image to a one-hot representation. In the design of Euler Net, developers included this extra network in the reasoning module.

---

> > ### Author Response · Authors · 2024-11-30
> >
> > Dear Reviewer,
> >
> > thanks again for giving us feedbacks. We uploaded a new version of paper. We would very much appreciate your continued feedback.

---

### Meta-Review · Area_Chair_Yajy · 2024-12-20

**Metareview:**

The paper studies whether current neural networks can perform syllogistic reasoning via Euler diagrams. The results indicate they fail, and the authors argue that neural networks need to go beyond vector embeddings to solve rigorous reasoning. In my opinion, the reviewers have provided a detailed and thoughtful assessment, presenting strong arguments regarding the paper's suitability for ICLR in its current state: weak presentation, unclear proof that transformers cannot do syllogistic reasoning, and unclear design choice for the neural architecture used, among others. These downsides should be addressed before publication. Please note that the overall judgment should not be taken as a statement regarding the usefulness of your research.

**Additional Comments On Reviewer Discussion:**

The discussion arose from problems and questions that had been raised in the reviews and also touched upon cognitive science aspects. One of the main topics was OOD and logical reasoning with neural networks. Some discussions were quite long but did not change the overall impression on the paper.

---

### Decision · Program_Chairs · 2025-01-22

Reject